# Evaluating the World Model Implicit in a Generative Model

**Keyon Vafa**
Harvard University

**Justin Y. Chen**
MIT

**Ashesh Rambachan**
MIT

**Jon Kleinberg**
Cornell University

**Sendhil Mullainathan**
MIT

## Abstract

Recent work suggests that large language models may implicitly learn world models. How should we assess this possibility? We formalize this question for the case where the underlying reality is governed by a deterministic finite automaton. This includes problems as diverse as simple logical reasoning, geographic navigation, game-playing, and chemistry. We propose new evaluation metrics for world model recovery inspired by the classic Myhill-Nerode theorem from language theory. We illustrate their utility in three domains: game playing, logic puzzles, and navigation. In all domains, the generative models we consider do well on existing diagnostics for assessing world models, but our evaluation metrics reveal their world models to be far less coherent than they appear. Such incoherence creates fragility: using a generative model to solve related but subtly different tasks can lead to failures. Building generative models that meaningfully capture the underlying logic of the domains they model would be immensely valuable; our results suggest new ways to assess how close a given model is to that goal.

## 1 Introduction

Large language models (LLMs) appear to have capacities that far exceed the next-token prediction task they were trained to perform [17, 39, 35]. Recent work suggests a reason: they are implicitly recovering high-fidelity representations of the underlying domains they are trained on [1, 20].

An algorithm that recovers a "world model" from sequence data would be extremely valuable. As an example, consider how one might build a navigation tool today: meticulously map each street and intersection, and then use a search algorithm to provide directions. The success of language models suggests an alternative approach: collect turn-by-turn sequences from trips in a city (e.g. "East North...") and then train a sequence model on them. If the sequence model successfully recovers the world model, we would obtain a map of the city without ever mapping it and a routing algorithm simply by predicting the next turn. This example is not far-fetched: it is the reason language models are used in scientific domains such as protein generation, genetics and chemistry [7, 21, 3, 14, 6].

All of this relies on the presumption that the sequence model has recovered the true world model; but how can we test whether it actually has? Answering this question requires first defining what we mean by the true world model. Toshniwal et al. [36] and Li et al. [20] proposed a concrete and influential approach: study whether sequence models trained on board game transcripts (e.g. chess and Othello) recover the underlying game rules. Inspired by this approach, we consider the case where the underlying world can be summarized by a finite collection of states and rules governing transitions between the states; this includes many domains such as logic [19], location tracking [28, 9], games [36, 20], and several of the scientific applications described above. As a result, the "world" in these domains can be modeled as a deterministic finite automaton (DFA).

38th Conference on Neural Information Processing Systems (NeurIPS 2024).

We show the difficulty in evaluating implicit world models. Consider an existing approach: for a given sequence, compare the next tokens outputted by the generative model to the set of valid next tokens for the state implied by that sequence [36, 20]. Though intuitive, this approach can fail to diagnose severe problems, and we illustrate this concretely. The classic Myhill-Nerode theorem [26, 27] provides intuition: every pair of distinct states can be distinguished by some sequence (admitted by one state but not the other). Unless those minimal distinguishing sequences are of length one, looking at the next *single* token outputted will not reliably assess whether the generative model has an accurate model of the underlying state.

The logic of Myhill-Nerode suggests two metrics for measuring whether a generative model effectively captures underlying states and transitions. The first metric summarizes *sequence compression*: under the DFA, sequences that lead to the same state must have the same continuations; so one can test whether the generative model has similar sequences of outputs when started on these two sequences. The second metric summarizes *sequence distinction*: under the DFA, two sequences that lead to distinct states should have distinct continuations; so one can test whether the generative model matches these distinct outputs when started at these two sequences. We formally define these metrics and provide model-agnostic procedures for calculating them when given query access to the true DFA.

To illustrate these ideas, we first take the stylized mapping example literally. We construct a turn-by-turn sequence dataset of taxi rides in New York City. We then assess to what extent transformers successfully recover the true street map of Manhattan. By the usual metrics, the transformers do very well: their predicted next-direction is a valid turn nearly 100% of the time and their state representations even appear to encode the current location of the ride. Our evaluation methods reveal they are very far from recovering the true street map of New York City. As a visualization, we use graph reconstruction techniques to recover each model's implicit street map of New York City. The resulting map bears little resemblance to the actual streets of Manhattan, containing streets with impossible physical orientations and flyovers above other streets (see Figure 3). Because these transformers fail to recover the true street map of New York City, they are fragile for downstream tasks. While they sometimes have amazing route planning abilities, their performance breaks down when detours are introduced.

These results are not unique to maps and navigation. For both Othello and logic puzzles, we use our evaluation metrics to show language models can perform remarkably well on some tasks despite being far from recovering the true world model. These results demonstrate the importance of using theoretically-grounded evaluation metrics if our goal is to build language models that capture accurate world models of the domains they are trained in. We release our benchmark dataset of taxi rides in New York City along with software implementing our evaluation metrics.[1]

**Related work.** Our paper builds on influential work studying whether generative models recover a world model in the context of games. Toshniwal et al. [36] and Li et al. [20] pioneered the study of games as a testbed for world model evaluation, studying tests for chess and Othello, respectively, which were further studied by Hazineh et al. [10] and Kuo et al. [18]. Our evaluation metrics apply to these games because they are DFAs. A common method for assessing whether a trained model has recovered a world model uses probes that assess whether a neural network's representation can recover some real-world state [11, 19, 1, 16, 20]. By contrast, our evaluation metrics are model-agnostic: they're based only on sequences. While the results from our evaluation metrics sometimes align with those used in existing work, they also reveal incoherence in world models that are not captured by existing diagnostics.

We study whether a language model trained on sequences of directions recovers the true underlying map. This question relates to other state tracking and navigation problems studied in the language modeling literature [31, 32]. For example, Patel & Pavlick [28] show that larger LLMs ground spatial concepts like cardinal directions to locations in a grid world and generalize to various grid layouts. Relatedly, Schumann & Riezler [30] demonstrate that transformer-based models can generate navigation instructions in language from underlying graphs. Additionally, Guan et al. [9] use LLMs to perform planning tasks from natural language descriptions. Our results suggest that LLMs can perform some of these tasks well (such as finding shortest paths between two points on a map) without having a coherent world model.

Additionally, our evaluation metrics compare the language accepted by a sequence model to that of an underlying DFA. Existing work studies whether transformers and other sequence models are theoretically capable of recognizing languages in different complexity classes [34, 4, 22, 23, 24].

---

[1] https://github.com/keyonvafa/world-model-evaluation

Most relevant to our work, Liu et al. [22] show that low-depth transformers can theoretically represent any finite state automata, and show that transformers trained explicitly to predict their labeled states are capable of doing so. In contrast, our paper doesn't aim to study whether models are theoretically capable of recovering underlying automata or whether they can do so when given state labels. Instead, we provide metrics for assessing how closely a given model recovers the underlying DFA.

## 2  Framework

In this section, we lay out a framework to interface between generative sequence models and world models represented by deterministic finite automata. Both of these are built on the shared scaffolding of tokens, sequences (a.k.a. strings), and languages.

**Tokens and sequences.**  We consider a finite alphabet $\Sigma$ with tokens $a \in \Sigma$, and sequences $s = (a_1, a_2, \ldots)$. Let $\Sigma^*$ denote the collection of sequences on the alphabet.

**Generative models.**  A *generative model* $m(\cdot)\colon \Sigma^* \to \Delta(\Sigma)$ is a probability distribution over next-tokens given an input sequence. That is, $m(s) \in \Delta(\Sigma)$, and $m(a \mid s)$ is the probability assigned to token $a \in \Sigma$ given an input sequence $s$. Starting at a sequence $s$, the set of non-empty sequences the model can generate with positive probability is:

$$L^m(s) = \{a_1 a_2 ... a_k : \forall j < k, \ m(a_{j+1} \mid s a_1 ... a_j) > 0\}.$$

For simplicity, we write the equation above for next-tokens with nonzero probability, but in practice we set a minimum probability $\epsilon > 0$ corresponding to next-tokens with non-negligible probability.

**Deterministic finite automata (DFA).**  We use standard notation for a deterministic finite state automaton $W = (Q, \Sigma, \delta, q_0, F)$ (see Appendix C for a complete definition). As a simplifying assumption, we consider the case where there is a special state $q_{\text{reject}}$ with no outgoing transitions and $F = Q \setminus \{q_{\text{reject}}\}$ (i.e., the DFA accepts all valid states). An extended transition function $\hat{\delta}$ takes a state and a sequence, and it inductively applies $\delta$ to each token of the sequence. A token or a sequence is *valid* if and only if the output of $\delta$ or $\hat{\delta}$ respectively starting from $q_0$ is not $q_{\text{reject}}$.

We define $L^W(q)$ to be the set of valid, non-empty sequences that are accepted by the DFA starting at state $q$. We also define $q(s) \in F$ to be the state that sequence $s$ leads to in the DFA starting from $q_0$ and $S(q) \subseteq \Sigma^*$ to be the collection of all sequences that lead from state $q_0$ to state $q$ in the DFA.

### 2.1  Recovering world models

Throughout this paper we assume that the ground-truth sequences used to train and test a generative model belong to the language of a deterministic finite state automaton $W$. This generalizes past work (e.g., on assuming sequences come from legal moves in a game [36, 20]) and allows us to formally define world recovery.

**Definition 2.1.**  A generative model $m(\cdot)$ **recovers the DFA** $W$ if

$$\forall q \in F, \forall s \in S(q)\colon L^W(q) = L^m(s).$$

That is, recovery requires that a sequence can be generated with positive probability by the model $m(\cdot)$ if and only if the sequence is valid in the DFA $W$.

Recovery is defined at the language level. However, generative models are often built and evaluated token-by-token. It turns out that exact next-token prediction is enough for recovery of the language of the world model.

**Definition 2.2.**  A generative model $m(\cdot)$ satisfies **exact next-token prediction** under the DFA $W$ if

$$\forall q \in F, \forall s \in S(q), \forall a \in \Sigma\colon m(a \mid s) > 0 \iff \delta(q, a) \neq q_{\text{reject}}.$$

**Proposition 2.3.**  *A generative model $m(\cdot)$ recovers the DFA $W$ if and only if it satisfies exact next-token prediction under the DFA $W$.*

Proposition 2.3 (proof given in Appendix A) suggests a way to evaluate whether a generative model recovers the true DFA: assess the validity of next-token predictions. Existing world model diagnostics are motivated by this intuition; for example, one way that Toshniwal et al. [36] and Li et al. [20] assess world model recovery is by measuring the percent of top next-token predictions that are valid.

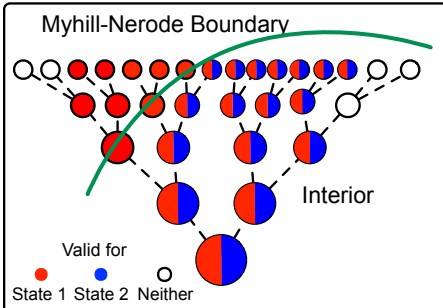
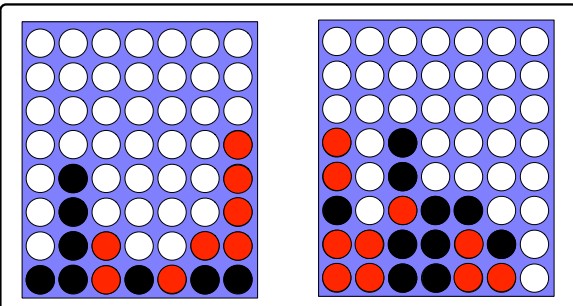

**Figure 1:** On the left, a visual depiction of a Myhill-Nerode boundary and interior. On the right, examples of two states for cumulative Connect-4. Both states have the same set of valid next moves. The shortest sequence in the Myhill-Nerode boundary has length 4, and the boundary contains sequences up to length 30. The interior contains approximately $8.8 \times 10^{27}$ sequences of length 29 that do not distinguish the two boards.

## 2.2 Next-token prediction is a fragile metric for recovering structure

Next-token prediction, however, is a limited evaluation metric. While exact next-token prediction implies perfect world model recovery, being very nearly correct on next-token prediction does not mean having very nearly recovered the world model. This can be illustrated by a simple example.

**Example: Cumulative Connect-4.** Consider a vertical grid with $n$ rows and 7 columns. Two players take turns dropping a disk in a column, and they can choose any column that contains less than $n$ disks. When a disk is dropped in a column, it occupies the bottom-most position that isn't occupied by another disk, and it remains in that position for the full game. The game continues until the entire board is filled, for $7n$ moves, regardless of whether a player has achieved four in a row. Games are represented as sequences of moves, where each sequence has $7n$ tokens and each token is an integer between 1 and 7 indicating the column the disk is placed in. Here, $\Sigma = \{1, \ldots, 7\}$ denotes the columns and the state corresponds to the count in each column. A column is a valid move if that column is not already filled.

Consider a generative model that outputs $\{1, \ldots, 7\}$ with uniform probability given any sequence, i.e. $m(a \mid s) = m(a' \mid s') = 1/7$ for all $a, a' \in \Sigma$ and $s, s' \in \Sigma^*$. This model clearly encodes no information about the board. However, for any board where there are no columns filled, this model provides a valid next move (e.g., the right panel of Figure 1), and so it will be a near-perfect next-token predictor when $n$ is large. For example, when $n = 1000$, it predicts a valid next move for more than 99% of all states. Metrics based on next-token prediction will imply this algorithm is close to recovering a world model.

## 2.3 The Myhill-Nerode interior and boundary

Cumulative Connect-4 points to a general fragility in next-token prediction as an evaluation metric that can be understood in the context of the Myhill-Nerode theorem [26, 27], a classic result from language theory. The Myhill-Nerode theorem states that the sets of sequences accepted by a minimal DFA starting at two distinct states are distinct (see Appendix C for a full statement). More formally, for states $q_1 \neq q_2$, we have $L^W(q_1) \neq L^W(q_2)$. However, while distinct, the two sets may exhibit a great deal of overlap. Cumulative Connect-4 exhibits this behavior; any board for which there are less than $k$ disks in each column will have the same set of valid moves for the next $n - k$ moves. This intuition motivates a pair of definitions:

**Definition 2.4.** Given a DFA $W$, the **Myhill-Nerode interior** for the pair $q_1, q_2 \in F$ is the set of sequences accepted when starting at both states:

$$\text{MNI}^W(q_1, q_2) = \{s \in \Sigma^* \mid s \in L^W(q_1) \cap L^W(q_2)\}.$$

The **Myhill-Nerode boundary** is the set of minimal suffixes accepted by a DFA at $q_1$ but not $q_2$:

$$\text{MNB}^W(q_1, q_2) = \{s = a_1 a_2 \ldots a_k \mid s \in L^W(q_1) \setminus L^W(q_2) \text{ and } \forall j < k : a_1 \ldots a_j \in \text{MNI}^W(q_1, q_2)\}.$$

Figure 1 depicts an example Myhill-Nerode interior and boundary for cumulative Connect 4. Sequences on the interior are accepted by both states; it is only when we reach the boundary that these states will be distinguishable. Thus, models that pool together states with large interiors will

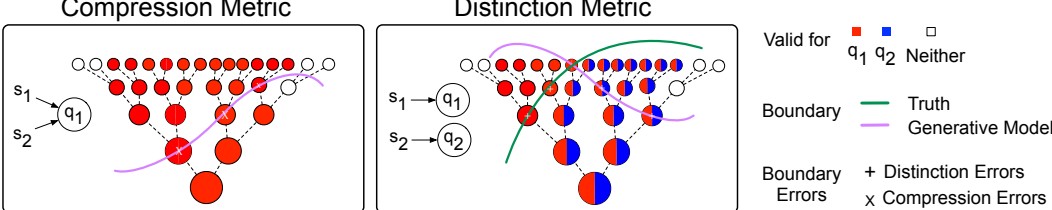

**Figure 2:** A visual depiction of our two evaluation metrics. A compression error is a model failing to recognize that two sequences that result in the same state should accept the same suffixes. A distinction error is a model failing to find the right distinguishing suffixes for two sequences that lead to different states. Our metrics measure errors at the boundary, which are visually depicted above.

perform well on next-token prediction tests; this is why the simple generative model succeeds in the cumulative Connect-4 example. To properly differentiate states, we must consider sequences that are long enough to be differentiated. In the remainder of the paper, we (i) use the Myhill-Nerode logic to develop new evaluation metrics and (ii) apply these to several applications.

### 2.4 Compression and distinction metrics for evaluating world models

We propose metrics to evaluate a model's implicit world model by comparing the true Myhill-Nerode boundary to the one implied by the model.

**Definition 2.5.** For two sequences $s_1, s_2$, the **Myhill-Nerode boundary implied by model** $m(\cdot)$ is

$$\mathrm{MNB}^m(s_1, s_2) = \{x = x_1...x_k \mid x \in L^m(s_1) \setminus L^m(s_2) \text{ and } \forall j < k : x_1...x_j \in L^m(s_1) \cap L^m(s_2)\}. \quad (1)$$

This is the set of minimal suffixes that are accepted by the model conditioned on $s_1$ but not $s_2$. Since we now focus on the generative model rather than the DFA, the definition refers to pairs of sequences rather than to pairs of states.

Our evaluation metrics summarize how well a generative model identifies sequences that distinguish a given pair of states. Given a pair of states $q_1$ and $q_2$, the metric is formed by first sampling sequences that lead to each state, $s_1 \in S(q_1)$ and $s_2 \in S(q_2)$. We then calculate the true Myhill-Nerode boundary between the states and the model's boundary between the sequences. Our metrics then compare the resulting boundaries using two statistics as building blocks:

**Definition 2.6.** The **boundary recall** of generative model $m(\cdot)$ with respect to a DFA $W$ is defined as

$$\frac{|\mathrm{MNB}^W(q_1, q_2) \cap (L^m(s_1) \setminus L^m(s_2))|}{|\mathrm{MNB}^W(q_1, q_2)|}, \quad (2)$$

and the **boundary precision** is defined as

$$\frac{|\mathrm{MNB}^m(s_1, s_2) \cap (L^W(q_1) \setminus L^W(q_2))|}{|\mathrm{MNB}^m(s_1, s_2)|}. \quad (3)$$

Notice that boundary recall and boundary precision are not affected by whether the Myhill-Nerode interior is large between the two states. Returning to cumulative Connect-4, the simple generative model that outputs $\{1, \ldots, 7\}$ with equal probability will perform poorly on these metrics; its recall will be 0 for all pairs of distinct states.

Based on the building blocks of recall and precision, we construct evaluation metrics to summarize whether the generative model correctly *compresses* sequences that arrive at the same state under the DFA and correctly *distinguishes* sequences that arrive at different states under the DFA. These two metrics correspond to different methods of sampling state pairs.

**Sequence compression metric.** To evaluate sequence compression, we sample equal state pairs $q_1 = q_2$. Since a DFA provides multiple ways to arrive at the same state, this test assesses whether a generative model recognizes that two sequences correspond to the same state. For example, in cumulative Connect-4, there may be multiple sequences that arrive at the same board position. Recall is undefined for equal states because there is no true boundary, so our compression metric only reports

precision, averaged over states sampled uniformly at random (we say a generative model's precision is 1 if its boundary is correctly empty).

**Sequence distinction metric.** To evaluate sequence distinction, we sample distinct state pairs, i.e. $q_1 \neq q_2$. Here, there must be a true boundary, so we test how well a generative model recovers it. We report both precision and recall averaged over state pairs sampled uniformly at random.

Both metrics are depicted in Figure 2. Although we have defined a generative model as accepting all sequences it assigns positive probability to, in practice sequence models are regularized to assign all sequences nonzero probability. Our evaluation metrics therefore depend on an acceptance threshold parameter $\epsilon > 0$. In practice, we explore sensitivity to different values of $\epsilon$ and other acceptance mechanisms. We present ablations and other details in more depth in Section 3 and Appendix E.

# 3 Illustration: Do Transformers Recover the Street Map of New York City?

To illustrate these metrics, we create a dataset consisting of taxi rides in New York City. We process each ride into sequences of turn-by-turn directions and train transformers to predict the next direction. We show that transformers trained on these sequences have surprising route planning abilities: they not only find valid routes between two intersections but usually find the shortest path.

We then examine the underlying world model of the trained models. Despite the route planning capabilities of these models, our metrics reveal that their underlying world models are incoherent. Using a graph reconstruction technique, we show that each model's implicit street map of New York City bears little resemblance to the actual map. Finally, we demonstrate that the route planning capabilities of these models break down when detours are introduced, a consequence of their incoherent world models.

## 3.1 Data and models

We base our analysis on a dataset of taxi rides released by the NYC Taxi & Limousine Commission, containing the latitude and longitude of each ride's pickup and dropoff location in Manhattan. Each taxi ride obeys a true world model: the weighted graph corresponding to the system of intersections and streets in New York City. The graph is defined as $G = (V, E, W)$, where $V$ is the set of intersections, $E$ the set of streets, and $W : E \to \mathbb{R}^+$ a weighting function containing the distance of each street.[2] Each edge is labeled corresponding to its cardinal direction, represented as a function $D : V \times V \to \{\square, \mathsf{N}, \mathsf{S}, \mathsf{E}, \mathsf{W}, \mathsf{NE}, \mathsf{NW}, \mathsf{SE}, \mathsf{SW}\}$ with $\square$ indicating that the edge does not exist. Each intersection has at most one edge in each direction. The graph has 4580 nodes (i.e. intersections) and 9846 edges (i.e. streets).

A traversal is a sequence of nodes where an edge exists between each consecutive node in the sequence. To study how the construction of traversals affects the resulting generative model, we consider three different approaches. *Shortest paths* constructs traversals by finding the shortest path between two nodes. Since these may not be reflective of real-world traversals due to traffic conditions, *noisy shortest paths* constructs multiple shortest paths by perturbing the magnitude of each edge weight in the underlying graph. Finally, *random walks* samples random traversals instead of approximating shortest paths. See Appendix F for details.

We convert each traversal into a sequence of directions. Each sequence begins with the origin and destination, followed by the cardinal directions in the traversal, and concludes with a special end-of-sequence token. Figure 5 gives an example of a set directions and the corresponding path. Since this language corresponds to a DFA $W$ with $|V|^2 + 1$ accept states, corresponding to all combinations of current intersection/destination intersection pairs and an additional end state, we can apply the evaluation metrics in Section 2.4.

We randomly split data into train and test splits, ensuring no origin-destination pair is in both train and test sets. We include all sequences containing less than 100 directions. Our training sets consist of 2.9M sequences (120M tokens) for shortest paths; 31M sequences (1.7B tokens) for noisy shortest paths; and 91M sequences (4.7B tokens) for random walks. We train two types of transformers [38] from scratch using next-token prediction for each dataset: an 89.3M parameter model consisting of 12 layers, 768 hidden dimensions, and 12 heads; and a 1.5B parameter model consisting of 48 layers, 1600 hidden dimensions, and 25 heads. We follow the architecture of GPT-2 for each model [29]. We train models on

---

[2]A real-world intersection may be represented as multiple intersections here. For example, if a turn is only valid from one direction, it is represented as two different nodes.

| | Existing metrics | | Proposed metrics | | |
|---|---|---|---|---|---|
| | Next-token test | Current state probe | Compression precision | Distinction precision | Distinction recall |
| Untrained transformer | 0.03 (0.00) | 0.10 (0.00) | 0.00 (0.00) | 0.00 (0.00) | 0.00 (0.00) |
| Shortest paths | 1.00 (0.00) | 0.91 (0.00) | 0.10 (0.01) | 0.35 (0.02) | 0.20 (0.01) |
| Noisy shortest paths | 1.00 (0.00) | 0.92 (0.00) | 0.05 (0.01) | 0.37 (0.02) | 0.24 (0.01) |
| Random walks | 1.00 (0.00) | 0.99 (0.00) | 0.50 (0.02) | 0.99 (0.00) | 1.00 (0.00) |
| True world model | 1.00 | — | 1.00 | 1.00 | 1.00 |

**Table 1:** Sequence compression and distinction metrics for world models compared to existing metrics (standard errors in parentheses). Models that do well on existing metrics can perform poorly on ours.

8 A100 GPUs. For each dataset, we analyze the model with the best held-out performance: the 89.3M parameter model for shortest paths, and the 1.5B parameter for noisy shortest paths and random walks.

## 3.2 Evaluating world models

To assess their capabilities, we first assess whether the trained models can recover the shortest paths between unseen (origin, destination) pairs. We prompt each model with (origin, destination) pairs from the test set and use greedy decoding to generate a set of directions. All models consistently generate valid traversals — between 96% and 99%. Impressively, 97% of the sequences generated by the shortest paths model are the true shortest path, and 94% of the sequences generated by the model trained on noisy shortest paths find a shortest path for one of the noisy graphs used to generate data. Figure 5 provides an example of a shortest path traversal.

To assess whether these capabilities correspond to coherent implicit world models, we first consider two existing diagnostics [36, 20]. The **next-token test** assesses whether a model, when conditioned on each subsequence in the test set, predicts a legal turn for its top-1 predicted next-token. In our example, a directional move is legal if a street in the direction exists at the current intersection. Predicting the end token is only legal if the traversal implied by the sequence is at the listed destination. Meanwhile, the **current-state probe** trains a probe [11] from a transformer's representation to predict the current intersection implied by the directions so far. We train a linear probe on a transformer's last layer representation.

To implement the sequence compression metric, we randomly sample states (i.e., [intersection, destination] pairs) and two distinct traversals (i.e. prefixes) that arrive at each state. We then assess whether a model correctly admits the same suffixes for each prefix. We average over pairs of prefixes to report a score for each state and average over states to report a final score. To implement the sequence distinction metrics, we sample pairs of distinct states and traversals (i.e. prefixes) that arrive at each state, comparing the model's approximate Myhill-Nerode boundary to the true one. We average over pairs of prefixes to report a score for each pair of states, and average over 1000 randomly sampled state pairs to report a final scores. Both metrics depend on a threshold parameter $\epsilon$: a prefix is only sampled or accepted if the model's assigned probability for each token is above $\epsilon$. Here, we consider $\epsilon = 0.01$ for all models and metrics. We describe implementation details, provide parameter ablations, and consider other acceptance rules (e.g. top-p and top-k) in Appendix E.

Table 1 summarizes our results. As references, we compare each trained transformer to a randomly initialized transformer baseline following Li et al. [20] as well as to the true world model. The three trained transformers perform exceptionally well on existing diagnostics; nearly 100% of next-token predictions are valid and the probe recovers the true intersection for more than 90% of examples.[3]

Our evaluation metrics, however, reveal that these existing diagnostics are incomplete. All trained transformers perform poorly on sequence compression, frequently failing to recognize that two prefixes leading to the same state should admit the same continuations. Even the transformer trained on random walks, which sees many distinct types of traversals during training, fails to compress prefixes for half the states. For the sequence distinction metrics, the transformers trained on shortest paths or noisy shortest paths perform poorly. In contrast, the transformer trained on random walks performs well on the sequence distinction metric. Both metrics are therefore valuable for evaluating world models; a model can perform well on one metric and poorly on the other. Here, a model that

---

[3]While the next-token test accuracy is rounded to 100%, no model performs perfectly.

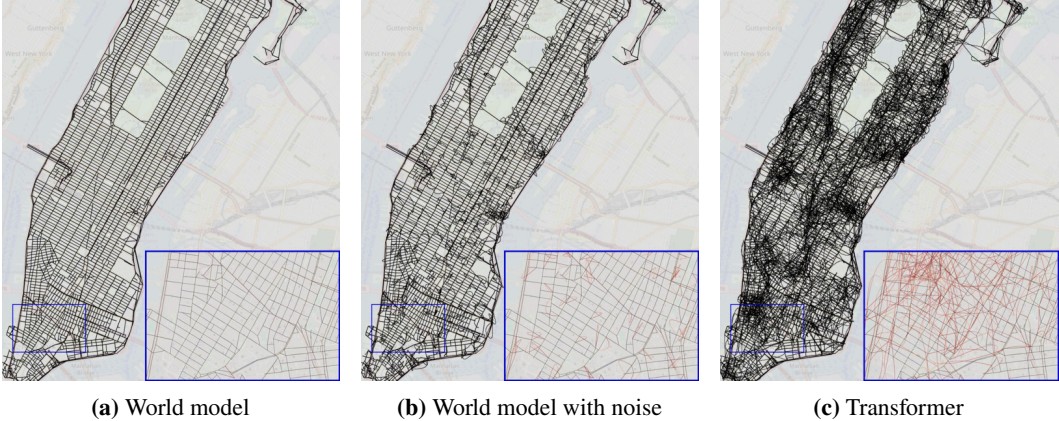

| **(a)** World model | **(b)** World model with noise | **(c)** Transformer |

**Figure 3:** Reconstructed maps of Manhattan from sequences produced by three models: the true world model (left), the true world model corrupted with noise (middle), and a transformer trained on random walks (right). Edges exit nodes in their specified cardinal direction. In the zoomed-in images, edges belonging to the true graph are black and false edges added by the reconstruction algorithm are red. We host interactive reconstructed maps from transformers at the following links: shortest paths, noisy shortest paths, and random walks.

distinguishes separate states at a high rate fails to recognize that two prefixes that lead to the same state should have the same valid continuations.

## 3.3 Reconstructing implicit maps

Our evaluation metrics point to deficiencies in recovering world models. We now show that these metrics reveal underlying incoherence. In the maps setting, the state structure of the true world model is easy to interpret and visualize: it is defined by the map itself. We attempt to "reconstruct" the map implied by sequences sampled from each generative model.

Reconstruction is an open-ended problem: the generative model produces directions between an origin and destination that do not necessarily correspond to a fixed graph over the intersections in Manhattan. To narrow the scope, our goal is to produce a visually interpretable reconstructed map. To that end, we fix the reconstructed graph to have the same set of vertices as the true world model, corresponding to intersections in Manhattan, and ensure that the reconstruction algorithm returns a map consistent with the true model whenever it is run on valid sequences. Further, (a) we enforce each node has at most one outgoing edge of any direction, (b) we limit the maximum degree of each node, and (c) we limit the Euclidean distance spanned by any edge. Altogether, our reconstruction algorithm gives the generative model the benefit of the doubt, attempting to reconstruct edges belonging to the true map until forced to do otherwise in order to map a generated sequence. The algorithm is detailed in Appendix B.

Figure 3 shows three reconstructed maps using sequences generated by the transformer trained on random walks. The sequences underlying each map are generated by randomly sampling 6400 (origin, destination) pairs and then sampling the model's traversal for each pair (Appendix G shows similar results for when the distribution of origin/destination pairs follows the sampling distribution used to train each model). On the left is the reconstructed map on only sequences which are valid under the true world model. On the right is the reconstructed map using the transformer's sequences. The transformer's underlying world model is incoherent; it recovers streets whose orientations are physically impossible (e.g. labeled NW but facing east) and require flyovers above other streets.

To show that this map is not the product of a model that has the right world model but makes a few transcription errors, we artificially corrupt sequences drawn from the true model. With probability equal to the probability of an error for the random walks transformer, we randomly re-label an edge in a sequence consistent with the world model. The middle panel of Figure 3 shows the reconstructed graph. It is much closer to the true world model than the transformer (which makes errors at the same rate). While these results are for random walks and one setting of graph reconstruction, Appendix G shows maps for the other models and different reconstruction settings. All settings recover incoherent underlying maps.

|  | | Probability of detour | | | |
|---|---|---|---|---|---|
|  | | 0% | 1% | 10% | 50% | 75% |
| **Random detours** | Shortest paths | 0.99 (0.01) | 0.69 (0.05) | 0.08 (0.03) | 0.00 (0.00) | 0.00 (0.00) |
|  | Noisy shortest paths | 0.96 (0.02) | 0.52 (0.05) | 0.03 (0.02) | 0.00 (0.00) | 0.00 (0.00) |
|  | Random walks | 0.99 (0.01) | 0.99 (0.01) | 1.00 (0.00) | 0.97 (0.02) | 0.74 (0.04) |
|  | True world model | 1.00 | 1.00 | 1.00 | 1.00 | 1.00 |
| **Adversarial detours** | Shortest paths | 0.99 (0.01) | 0.66 (0.05) | 0.06 (0.02) | 0.00 (0.00) | 0.00 (0.00) |
|  | Noisy shortest paths | 0.96 (0.02) | 0.64 (0.05) | 0.04 (0.02) | 0.00 (0.00) | 0.00 (0.00) |
|  | Random walks | 0.99 (0.01) | 1.00 (0.00) | 1.00 (0.00) | 0.93 (0.03) | 0.51 (0.05) |
|  | True world model | 1.00 | 1.00 | 1.00 | 1.00 | 1.00 |

**Table 2:** The fraction of traversals that are valid when detours are introduced (standard errors in parentheses).

### 3.4 Implication of failing to recover the world model: detour fragility

Does it matter that the transformer has an incoherent world model? After all, it does very well at the practical task of finding shortest paths. Here we look at a slightly adjacent task and consider a driver facing detours while driving; how well does the model re-route?

Concretely, we feed each transformer an (origin, destination) pair from the test set and greedily decode a traversal. But with probability $p$ for each token, we add one of two kinds of detours: for "random detours", the model's proposed token is replaced with a randomly chosen (true) valid token; for "adversarial detours", it is replaced with the model's lowest ranked valid token. We always ensure a valid path to the destination exists (shorter than length 100) after each detour. Table 2 shows the fraction of valid traversals produced. While all models perform well initially, detours erode performance, illustrating how faulty world models can prove problematic. Notably, the transformer trained on random walks is more robust to detours than models trained on (noisy) shortest paths, mirroring its advantage on our proposed evaluation metrics. The similarity between model performance on these evaluation metrics and detour robustness illustrates the effectiveness of our proposed metrics for assessing world model recovery.

## 4 Other Applications: Othello and Logic Puzzles

We apply our evaluation metrics to two other settings: sequence models trained on games of Othello and large language models prompted to solve logic puzzles. In both cases, our framework finds the same type of incoherence that we found in the previous section for maps.

**Othello.** Li et al. [20] study the question of evaluating world models in the context of Othello, a board game that consists of players placing tokens on an 8x8 board. They train transformers on game transcripts to predict the next move of each game. They show these models perform well on both the next-token test and current-state probe considered in Section 2.4. Since the true Othello game can be represented as a DFA, we can apply our world model evaluation metrics. The sequence compression metric assesses whether openings that lead to the same board position have the same predicted next moves, while the sequence distinction metrics assess whether the model can differentiate two distinct boards.

We apply our metrics to the two Othello sequence models considered by Li et al. [20]: one trained on real games from Othello championship tournaments and another trained on synthetic games. Table 3 in Appendix D reports the metrics in both settings. The model trained on real games performs poorly on both compression and distinction metrics, failing to group together most pairs of game openings that lead to the same board. In contrast, the model trained on synthetic games performs well on both metrics. This discernment is not captured by the existing metrics, which show both models performing similarly. We validate this discernment by performing a "detours" exercise for Othello in Table 4 in Appendix D; while the model trained on synthetic data produces near-perfect games regardless of detours, the model trained on championship data fails immediately. Similar to the navigation setting, we again find that models trained on random/synthetic data recover more structure than those trained on real-world data.

**Logic puzzles.** We consider an additional application involving LLMs. Our metrics require that the ground truth language can be expressed as a DFA, so we consider a "seating arrangement" logic puzzle similar to those in Suzgun et al. [35]. There are $n$ seats and $n$ individuals. The vocabulary consists of statements like "Person 'A' is sitting in seat 1" and "Person 'B' is two seats away from

| | Capabilities | Proposed metrics | |
|---|---|---|---|
| | Task accuracy | Compression precision | Distinction recall |
| Llama-2 (70B) | 0.77 (0.03) | 0.08 (0.03) | 0.42 (0.04) |
| Llama-3 (8B) | 0.85 (0.02) | 0.18 (0.04) | 0.23 (0.03) |
| Llama-3 (70B) | 0.98 (0.00) | 0.25 (0.04) | 0.57 (0.04) |
| Mixtral-8x22B | 0.88 (0.01) | 0.35 (0.05) | 0.57 (0.05) |
| Qwen 1.5 (72B) | 0.88 (0.02) | 0.21 (0.04) | 0.56 (0.03) |
| Qwen 1.5 (110B) | 0.98 (0.00) | 0.53 (0.05) | 0.53 (0.04) |
| GPT-3.5 (turbo) | 0.83 (0.02) | 0.33 (0.05) | 0.18 (0.03) |
| GPT-4 | 1.00 (0.00) | 0.21 (0.04) | 0.56 (0.03) |
| True world model | 1.00 | 1.00 | 1.00 |

*Example task prompt*

There are 3 individuals named A, B, and C, and there are 3 seats, positioned 1-3. We have the following statements:
 1. B is in seat 3
 2. B is 1 seat away from A
Based on this information, where is C seated? You can use chain-of-thought reasoning.

**Figure 4:** On the left, an example given to large language models to assess task capabilities. On the right, each model's average task performance along with their results on our proposed metrics. Models are very capable of solving logic puzzles despite not having a coherent world model.

Person 'C". A state is the set of seating arrangements that are consistent with all of the statements so far, and a statement is valid if it doesn't contradict all arrangements in the given state.

We analyze Llama 2 (70B) [37], Llama-3 (8B and 70B), Mixtral (8x22B Instruct) [15], Qwen 1.5 Chat (72B and 110B) [2], GPT-3.5 turbo, and GPT-4. We consider $n$=3 individuals. We first assess whether the LLMs solve the logic puzzle task when the seating arrangement is fully specified by the statements. Figure 4 shows that most LLMs perform well at this task; GPT-4 is accurate on all examples. We then apply our metrics, assessing if each LLM compresses correctly (whether two sets of statements that lead to the same state lead to the same assessments) and has high recall for distinction (we do not compute precision for distinction because it is too expensive to approximate each LLM's Myhill-Nerode boundary). See Appendix E for further discussion.

Figure 4 shows the results averaged over 100 samples. While most LLMs can solve the logic puzzle when it's fully specified, they perform poorly on the compression and distinction metrics: no model has a compression precision higher than 40%. More than half the time a model is conditioned on two sequences with the same set of viable states, it asserts that different continuations are allowed for each sequence; see Figure 8 for an example. No model has distinction recall higher than 0.60. These results bring up an interesting point: LLMs can perform well at some logic tasks (such as when the seating arrangement is fully specified) without having a coherent world model.

## 5   Conclusion

In order to build high-fidelity algorithms that meaningfully capture the logic of the problems they model, we need ways to measure how close we are to that goal. This paper suggests theoretically grounded metrics for assessing the world models implicit inside generative models. Applications to maps, games, and logic puzzles suggest these metrics are both feasible to implement and insightful. Our results show that generative models can perform impressive tasks with incoherent world models (e.g. provide directions for taxi rides). But this incoherence makes them fragile for other tasks (e.g. providing directions when there are detours). This incoherence is also problematic when we hope to use a generative model to learn something latent about the world in scientific domains.

Our primary limitation is the focus on DFAs. While it is suitable for many applications like games, logic, and state tracking, extending it would be quite valuable, e.g. to situations where the underlying world model is more complicated than a DFA or is unknown. We suspect that the core ideas related to sequence compression and sequence distinction generalize to these richer settings, but leave that to future work.

## Acknowledgements

Keyon Vafa is supported by the Harvard Data Science Initiative. Justin Chen is supported by an NSF Graduate Research Fellowship under Grant No. 174530. Jon Kleinberg is supported in part by a Vannevar Bush Faculty Fellowship, a Simons Collaboration grant, and a grant from the MacArthur Foundation. We thank the Chicago Booth School of Business for generous support. We thank

Foundry[4] for providing the compute required to conduct this research. We also thank Sarah Bentley, Juan Carlos Perdomo, and Neekon Vafa for helpful comments and feedback.

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

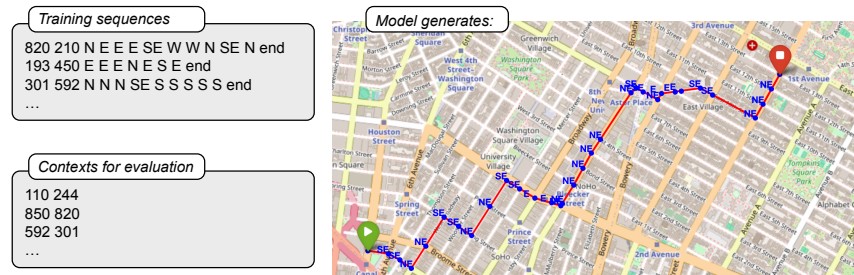

**Figure 5:** Examples of data and traversals. On the left are examples of sequences seen during training and contexts used for evaluation. On the right is an example traversal generated by a transformer trained on shortest paths data

# A  Proof

*Proof of Proposition 2.3.* We will first prove the forward direction that if a generative model $m(\cdot)$ recovers the world model DFA $W$, then $m(\cdot)$ satisfies exact next-token prediction. By assumption, $L^W(q) = L^m(s)$. Consider any state $q \in F$ and sequence reaching that state $s \in S(q)$. This implies that for any sequence $sa$ for any character $a \in \Sigma$,

$$\delta(q, a) \neq q_{\text{reject}} \iff m(a \mid s) > 0,$$

which is the definition of next-token prediction.

Now, we will prove the backwards direction that if $m(\cdot)$ achieves exact next-token prediction, it recovers $W$. Fix any state $q$ and sequence $s \in S(q)$. Consider any sequence $a_1 a_2 .. a_k \in \Sigma^*$ and let $q' = \hat{\delta}(q, a_1 a_2 .. a_k)$ be the state reached by following the sequence from $q$. Note that by definition of $q_{\text{reject}}$ not having any outgoing transitions,

$$q' \neq q_{\text{reject}} \iff (\forall j < k) : \hat{\delta}(q, a_1 a_2 .. a_j) \neq q_{\text{reject}}.$$

By assumption,

$$m(a_k \mid a_1 a_2 .. a_{k-1}) > 0 \iff q \neq q_{\text{reject}}.$$

If $a_1 a_2 ... a_k \in L^W(q)$, then $q' \neq q_{\text{reject}}$. It then must be the case that $m(a_j | a_1 a_2 .. a_{j-1}) > 0$ for all $j < k$, implying that $a_1 a_2 ... a_k \in L^m(s)$.

Conversely, if $a_1 a_2 ... a_k \notin L^W(q)$, then $q' = q_{\text{reject}}$. It follows that $m(a_k | a_1 a_2 .. a_{k-1}) = 0$, and thus $a_1 a_2 ... a_k \notin L^m(s)$. $\qquad\square$

# B  Reconstructed Maps

In this section, we give more details on our reconstruction algorithm and display maps for sequences generated from models trained on shortest paths, noisy shortest paths, and random walks each for several parameter settings.

## B.1  Algorithm

Our reconstruction algorithm in Algorithm 1 takes a set of sequences from a generative model and attempts to reconstruct the underlying map implied by the sequences. The reconstructed map has the same set of vertices as the true world model (i.e. the intersections of Manhattan), and we visualize each intersection by placing them at their real-world latitude/longitude. The reconstruction algorithm thus attempts to recover the edges and edge directions implied by each set of sequences.

For each sequence that is valid under the true map, the algorithm adds the true edges and edge directions implied by the sequence. In a sense, this algorithm gives the generative model the benefit of the doubt. However, sometimes the model errs, i.e. it produces a sequence which is only valid by adding an additional edge to the true map. In this case, our reconstruction algorithm adds a new edge at the first invalid step of the traversal. There are usually multiple possible edges to add that would make the traversal valid; the algorithm adds the edge that maximizes the number of subsequent steps

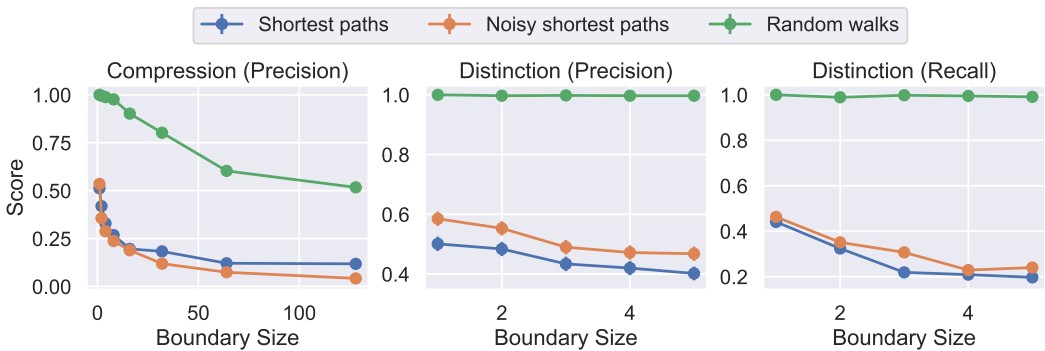

**Figure 6:** Performance metrics as a function of the maximum suffix length used to approximate the Myhill-Nerode boundary.

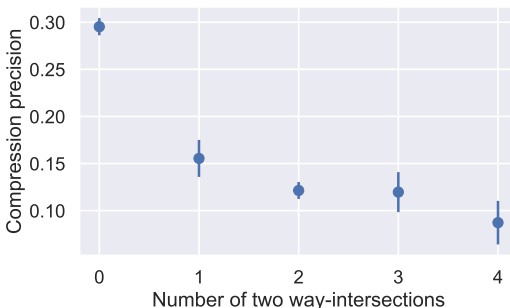

**Figure 7:** Compression precision worsens as there are more two-way streets at the current intersection. Plotted results are for the shortest paths model.

that would be valid. In other words, the algorithm adds new edges only when there is a discrepancy between the transformer and the world model, choosing the edge which greedily maximizes the number of subsequent steps that accord with the true world model. The direction label for each added edge is thus the token at the corresponding invalid step of the traversal.

Our algorithm may fail to reconstruct some of the input sequences as they cannot be made consistent with the partial graph we have reconstructed so far. Note that any reconstruction algorithm that satisfies the constraints in the main text of having a single edge with a given direction coming out of any intersection, a maximum degree, and a maximum edge distance will not be able to reconstruct every set of sequences. For example, two sequences $[v_1, v_2, \mathsf{E}, \mathsf{end}]$ and $[v_1, v_3, \mathsf{E}, \mathsf{end}]$ cannot be reconstructed without violating the first constraint.

### B.2   Maps

We include reconstructed maps built from 6400 transformer-generated sequences for transformers trained on shortest paths, noisy shortest paths (modeling traffic), and random walks in Manhattan. Each sequence is generated by randomly sampling an (origin, destination) pair and then sampling the model's traversal for each pair. Because the transformer never sees sequences of length more than 100 during training, we only sample pairs for which there exists a valid traversal in less than 100 moves. We note that this distribution of (origin, destination) pairs varies from the distributions used to train and evaluate each model, and the percent of traversals that are valid falls from >95% to 65-80%; however, Figure 12 shows similar maps when each map is constructed by sampling (origin, destination) pairs from the training/evaluation distributions. We vary the constrained maximum degree between 4 and 8 (the true map has maximum degree 4 and there are 8 possible cardinal directions), and the maximum edge distance between 1/2 and 1 mile.

Each map depicts the map of Manhattan implied by the graph reconstruction algorithm. Reconstructing maps involves adding edges between two intersections. We make sure the edges visually accord with the labels reconstructed by the algorithm. For example, if intersection $v_1$ is north of intersection $v_2$ in

**Algorithm 1** Graph Reconstruction from Sequences

**Input:** A set of sequences $\{s_i\}_{i=1}^n$ each consisting of source, destination, cardinal directions, and an end token; a true graph described by vertices $V \subset \mathbb{R}^2$ and edges $D : V \times V \to \{\square, \text{N}, \text{S}, \text{E}, \text{W}, \text{NE}, \text{NW}, \text{SE}, \text{SW}\}$; a maximum degree $deg$; and a maximum distance $dist$.
**Output:** Reconstructed edges $\hat{D}$.

---

1: Initialize $\hat{D}(u, v) \leftarrow \square$ for all $u, v \in V$
2: **for** $i \in [n]$ **do**
3:   $j \leftarrow 2$ is an index into the list of directions
4:   $u \leftarrow s_i[0]$ is the current node, starting at the source
5:   **while** $s_i[j] \neq$ end **do**
6:     **if** $\nexists v \in V$ s.t. $\hat{D}(u, v) = s_i[j]$ **then**
7:       **if** $|\{v \in V : \hat{D}(u, v) \neq \square\}| \geq deg$ **then**
8:         FAIL to reconstruct sequence and continue to line 2        // No room for new edges
9:       **else if** $\exists v \in V$ s.t. $D(u, v) = s_i[j]$ **then**
10:         $\hat{D}(u, v) \leftarrow s_i[j]$                     // Add an edge from the real graph
11:       **else**
12:         $N(u) \leftarrow \{v \in V : u, v$ within Euclidean distance $dist\}$
13:         $v' \leftarrow \arg\max_{v \in N(u)}$ # of steps through the while loop if $\hat{D}(u, v) = s_i[j]$ before reaching lines 8 or 13
14:         $\hat{D}(u, v) \leftarrow s_i[j]$              // Add new edge that gives longest continuation
15:     $j \leftarrow j + 1$
16:   **if** $u \neq s_i[1]$ **then**
17:     FAIL to construct sequence
18: **return** $\hat{D}$

---

the true map but the reconstruction algorithm recovers an edge from $v_2$ to $v_1$ labeled "South", we draw an edge that leaves $v_2$ facing south and loops back to $v_1$. In the zoomed-in images, edges belonging to the true map are in black and false edges added by the reconstruction algorithm are in red.

In each caption, we list the number of sequences the algorithm failed to reconstruct. In addition to the reconstructed map from all of the transformer's sequences, we plot the reconstructed map built only on sequences which are valid under the true world model as well as a reconstruction of those valid sequences with some artificial corruptions. For the artificial corruptions, each sequence is chosen to be corrupted with a fixed probability of 25% for shortest paths, 35% for noisy shortest paths, and 20% for random walks. These percentages correspond to the fraction of the transformer's sequences that are invalid, so the (b) and (c) subfigures of each plot have the same proportion of valid and invalid sequences.

We note that the maps corresponding to the random walk model visually have more edges than, for instance, the maps built from shortest paths data. This is due to the fact that while the inputs to the reconstruction algorithm have the same number of sequences, the length of the sequences differ between the different generative models. The total number of directions contained in the shortest path sequences is 983,182, for noisy shortest paths is 1,140,487, and for random walks is 1,584,549.

## C  Deterministic Finite Automata and Myhill-Nerode

Recall that we use a standard parameterization of a DFA as $W = (Q, \Sigma, \delta, q_0, F)$ (see [33]) with

1. $Q$ is a finite set of states,
2. $\Sigma$ is a finite set of characters,
3. $\delta : Q \times \Sigma \to Q$ is the transition function mapping a state and character to the next state,
4. $q_0 \in Q$ is the start state,
5. $F \subseteq Q$ is the set of accepting states.

In the rest of this section, we state the Myhill-Nerode theorem, which is the conceptual basis for our world-model test.

|                        | Compression precision | Distinction precision | Distinction recall |
| ---------------------- | :-------------------: | :-------------------: | :----------------: |
| Untrained transformer  | 0.00 (0.00)           | 0.02 (0.00)           | 0.14 (0.01)        |
| Championship Othello   | 0.00 (0.00)           | 0.65 (0.01)           | 0.27 (0.01)        |
| Synthetic Othello      | 0.98 (0.00)           | 0.99 (0.00)           | 1.00 (0.00)        |
| True world model       | 1.00                  | 1.00                  | 1.00               |

**Table 3:** The metrics from Section 2.4 applied to Othello sequence models. The championship and synthetic models refer to models trained on real-world tournament games and synthetic games, respectively.

**Definition C.1** (Equivalent sequences). Two sequences $s_1, s_2 \in \Sigma^*$ are called **equivalent** under a language $L$ if for all suffixes $x \in \Sigma^*$, $s_1 x \in L \iff s_2 x \in L$. This equivalence relation can be used to partition all strings into equivalence classes.

**Theorem C.2** ([26, 27]). *A language $L$ is regular if and only if it has a finite number of equivalence classes. Then, the minimal DFA accepting $L$ has a number of states equal to the number of classes. In the minimal DFA, for every pair of distinct states $q_1 \neq q_2$, there exists a suffix $x$ such that exactly one of $\hat{\delta}(q_1, x)$ or $\hat{\delta}(q_2, x)$ is in the set of accepting states $F$.*

# D   Additional results

Figure 5 shows an example of sequences and traversals in our dataset. Each training sequences consists of an origin node, a destination node, and a set of directions followed by an end node. To evaluate, we condition on (origin, destination) pairs that are unseen during training and generate a traversal from the model.

Figure 6 shows how our performance metrics vary as we consider different suffix lengths for approximating the Myhill-Nerode boundary. For compression precision, considering a boundary of size $k$ corresponds to sampling suffixes of length-$k$ for each suffix and measuring whether prefixes with the same state have the same length-$k$ suffixes. For distinction precision, we consider a boundary of size $k$ by only sampling $k$-length suffixes to approximate a model's Myhill-Nerode boundary. For distinction recall, we consider a boundary of size $k$ by only constructing the true Myhill-Nerode boundary based on $k$-length suffixes. The results in Figure 6 show the importance of considering larger boundaries as opposed to smaller ones (e.g. single next-tokens); for example, while the model trained on random walks scores 100% on compression precision when boundaries of length-1 are considered, its performance is 50% when the full Myhill-Nerode boundary is considered.

We also performed some analysis to explore why the models uniformly performed badly on compression tests in the maps setting. We found a negative correlation between the number of two-way streets at an intersection and the compression precision; as the number of two-way streets at an intersection increases, the model's ability to recognize that two sequences that lead to the same intersection are indeed in the same state worsens. Figure 7 plots this relationship for the shortest paths model.

Table 3 reports our metrics on models trained to play Othello. We use the transformer model checkpoints provided by Li et al. [20]. We perform 1000 samples of each test. Our metrics find that while the model trained on synthetic data recovers the true world model, the model trained on championship data does not. In Table 4, we perform a detour exercise analogous to the one performed for taxi rides in Section 3, where here a model's predicted move is replaced with another legal one. The detour results support the discernment between models found by our metrics; while the model trained on synthetic data produces near-perfect games regardless of detours, the model trained on championship data fails immediately.

# E   Evaluation metric details and ablations

Here we provide implementation details and ablations for the test described in Section 2.4 and implemented in Section 3. We discuss how it's implemented in each of the three settings — maps, Othello, and logic puzzles — and then show ablations with different settings.

|  |  | Probability of detour | | | | |
| --- | --- | --- | --- | --- | --- | --- |
|  |  | 0% | 1% | 10% | 25% | 50% |
| **Random detours** | Championship Othello | 1.00 (0.00) | 0.66 (0.05) | 0.05 (0.02) | 0.01 (0.01) | 0.01 (0.01) |
|  | Synthetic Othello | 1.00 (0.00) | 0.99 (0.01) | 0.97 (0.02) | 0.97 (0.02) | 0.99 (0.01) |
|  | True world model | 1.00 | 1.00 | 1.00 | 1.00 | 1.00 |
| **Adversarial detours** | Championship Othello | 1.00 (0.00) | 0.70 (0.05) | 0.01 (0.01) | 0.01 (0.01) | 0.00 (0.00) |
|  | Synthetic Othello | 1.00 (0.00) | 0.98 (0.01) | 0.99 (0.01) | 0.96 (0.02) | 0.97 (0.02) |
|  | True world model | 1.00 | 1.00 | 1.00 | 1.00 | 1.00 |

**Table 4:** The fraction of traversals that are valid when detours are introduced in Othello. For "random detours", a model's proposed token is replaced with a randomly chosen (true) valid token; for "adversarial detours", it is replaced with the model's lowest ranked valid token.

### E.1 Implementation details

**Maps.** For the compression test, we sample a state at random from all possible states, where each state is a (current intersection, destination intersection) tuple. We then sample two distinct sequences that lead to the same state. Because models are only trained on sequences of length 100 or less, we only sample sequences that are short enough to be possible to arrive at the destination in less than 100 moves. Among all possible sequences, we first sample a length $l$ uniformly at random, and then perform two random walks for $l$ steps in the reversed graph. This provides two distinct length-$l$ prefixes that lead to the same state, $s_1$ and $s_2$.

Because the true world's Myhill-Nerode boundary is empty for the compression test, we only need to compute the model's boundary. However, computing the model's boundary is intractable; it involves evaluating a transformer on exponentially many outputs. However, we don't actually need to compute the full boundary; the precision is 0 any time there's one sequence accepted by one prefix and not the other. So we approximate precision by Monte-Carlo sampling. We sample $M$ complete sequences from the model conditioned $s_1$, insuring that each token has higher than $\epsilon$ probability. We then check if each token in the sequence has higher than $\epsilon$ probability when the model is conditioned on $s_2$. If there exists a single violating sample, it means that the model has failed to compress the two prefixes. Therefore, sampling results in an upper-bound on performance. In practice, we use $M = 30$ samples. Below, we show that results are not very sensitive to the number of samples.

For the distinction test, we sample two distinct states, $q_1$ and $q_2$, uniformly at random, and sample sequences that lead to each state, $s_1$ and $s_2$, as before. As before, computing full Myhill-Nerode boundaries is intractable, so we approximate them. To approximate the true world model's Myhill-Nerode boundary, we consider all continuations of length $k$, and find the set of minimal suffixes that are accepted after $q_1$ but not $q_2$. To see which elements in the true boundary are distinguished by the model, we evaluate the model on each element in conditioned on $s_1$ vs $s_2$. We use $k = 5$, and find that results are robust across different $k \geq 5$. We again approximate the transformer's Myhill-Nerode boundary by taking $M = 30$ Monte-Carlo samples. For each sequence that's accepted after $s_1$ but not $s_2$, we find the minimal distinguishing suffix and include it in the boundary set. We then assess precision by calculating which elements in the model's boundary are acceptable after $q_1$ but not $q_2$.

For both tests, we get state-level scores by averaging the results over prefixes that lead to each state, and we report overall scores by averaging all sampled states.

**Othello.** The compression test for Othello involves sampling a board and then two sequences that lead to the board. We approximate this sampling by simulating 1000 random games and sampling a board at random from the set of unique boards that are visited by at least two unique games. This sampling provides us with two different sequences, $s_1$ and $s_2$, that lead to the same board, $q$. Like the above, we don't need to compute the model's full Myhill-Nerode boundary since the precision is 0 any time one sequence is accepted by $s_1$ and not $s_2$. Therefore, we again approximate precision with $M = 30$ Monte-Carlo samples, following the same method as performed for maps.

For the distinction test, we sample two distinct states $q_1$ and $q_2$ from the set of sampled games with the same length. We sample prefixes $s_1$ and $s_2$ from the empirical distribution of observed games. We approximate the transformer's Myhill-Nerode boundary in the same way as we did for maps, by taking $M$ Monte-Carlo samples of complete game trajectories. We perform the analogous sampling

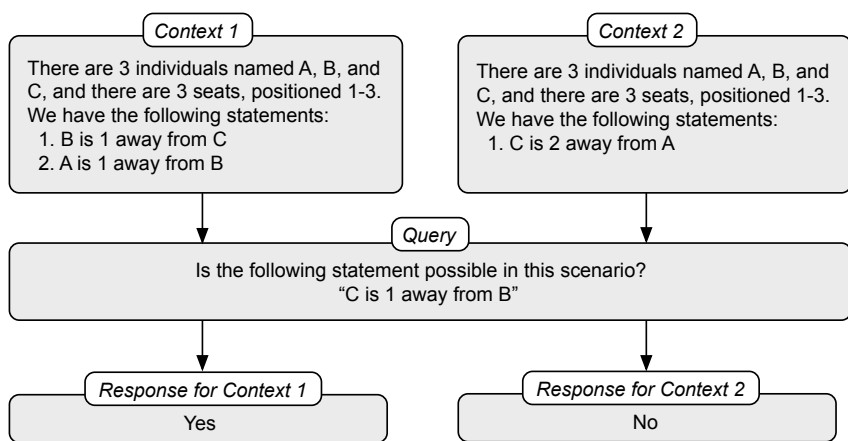

**Figure 8:** An example of a compression error for GPT-4 on the logic puzzle test. The model is prompted with statements that correspond to the same underlying state and a sample continuation. It assesses that the continuation is valid for one state yet invalid for the other.

technique for the true world model, sampling $M$ complete gameplay trajectories at random over the set of valid continuations. We use $M = 30$ for both sampling procedures and a threshold of $\epsilon = 0.01$.

**Logic puzzles.** Performing our test on large language models presents a challenge that we do not have token-level probability access. Moreover, because we allow large language models to perform chain-of-thought reasoning, it's computationally intractable to sample long continuations by marginalizing over the possible chain-of-thoughts.

Our test design is therefore based on prompting. See Figure 8 for an example. For the compression metric, we sample up to 2 statements uniformly at random from the set of possible statements to arrive at a state $q$. We then sample two prefixes $s_1$ and $s_2$ that lead to $q$ by repeatedly sampling the set of statements consistent with $q$ that don't narrow down the state space until the state implied by the statements is exactly that of $q$. The compression metric is failed for each state for which we can find a suffix that is accepted by the model prompted with one sequence but not the other. We sample 5 different possible continuation statements, half the time from the set of valid statements, half the time uniformly at random. We note that due to the nature of the compression metric, our reported metric is an overestimate of true capability. So the fact that no model performs above 0.35 with only 5 samples suggests heavy compression failure. We sample 100 states. See Figure 8 for an example of a compression error.

For the distinction metric, we again sample states $q_1 \neq q_2$ and sample sequences that lead to the states $s_1$ and $s_2$ using the same method as before. Because of the limited state space in our example, we can compute the true Myhill-Nerode boundary tractably. To test recall, we then only need to assess whether statements in the true boundary are accepted when the LLM is prompted by $s_1$ or $s_2$. Although the true Myhill-Nerode boundary is tractable to compute, it is still expensive to query an LLM with each example. Instead, we perform Monte Carlo sampling using $M$ statements from the true boundary to prompt the model. We consider $M = 5$ in our experiments.

We use the OpenAI API to query the GPT models, and use the Together AI API for all other LLMs. We prompt LLMs to perform chain-of-thought reasoning [40] for each query and automatically evaluate answers by prompting each model to output its response with the keyword "Answer:" followed by its answer. All queries are performed with greedy decoding.

## E.2 Ablations

Our test involves a few parameters, such as $\epsilon$ (the probability threshold for each model), the maximum suffix length $k$ used to approximate the true Myhill-Nerode boundary, and the number of Monte Carlo samples $m$ used to approximate the model's Myhill-Nerode boundary.

We begin by considering $\epsilon$, which dictates a tradeoff between precision and recall. In the main text, we consider $\epsilon = 0.01$. Table 5 reports results for other values of $\epsilon$ on the maps metrics. Empirically

|  |  | Compression precision | Distinction precision | Distinction recall |
|---|---|---|---|---|
| $\epsilon = 0.10$ | Shortest paths | 0.08 (0.03) | 0.30 (0.05) | 0.16 (0.03) |
|  | Noisy shortest paths | 0.04 (0.02) | 0.36 (0.05) | 0.20 (0.03) |
|  | Random walks | 0.16 (0.04) | 1.00 (0.00) | 1.00 (0.00) |
| $\epsilon = 10^{-2}$ | Shortest paths | 0.10 (0.03) | 0.38 (0.05) | 0.18 (0.03) |
|  | Noisy shortest paths | 0.08 (0.03) | 0.38 (0.05) | 0.26 (0.03) |
|  | Random walks | 0.46 (0.06) | 1.00 (0.00) | 1.00 (0.00) |
| $\epsilon = 10^{-4}$ | Shortest paths | 0.36 (0.05) | 0.29 (0.05) | 0.29 (0.04) |
|  | Noisy shortest paths | 0.29 (0.05) | 0.35 (0.05) | 0.34 (0.04) |
|  | Random walks | 0.79 (0.05) | 1.00 (0.00) | 0.97 (0.02) |
| $\epsilon = 10^{-6}$ | Shortest paths | 0.70 (0.05) | 0.29 (0.05) | 0.29 (0.04) |
|  | Noisy shortest paths | 0.70 (0.05) | 0.48 (0.05) | 0.23 (0.04) |
|  | Random walks | 0.99 (0.01) | 0.99 (0.01) | 0.11 (0.03) |

**Table 5:** Compression and distinction test results for different values of $\epsilon$ for models trained on map data.

|  |  | Compression precision | Distinction precision | Distinction recall |
|---|---|---|---|---|
| $k = 1$ | Shortest paths | 0.14 (0.04) | 0.33 (0.05) | 0.17 (0.03) |
|  | Noisy shortest paths | 0.06 (0.04) | 0.35 (0.07) | 0.11 (0.03) |
|  | Random walks | 0.40 (0.08) | 0.69 (0.06) | 0.30 (0.04) |
| $k = 2$ | Shortest paths | 0.21 (0.05) | 0.32 (0.05) | 0.17 (0.03) |
|  | Noisy shortest paths | 0.07 (0.04) | 0.31 (0.07) | 0.23 (0.05) |
|  | Random walks | 0.21 (0.07) | 0.93 (0.03) | 0.73 (0.05) |
| $k = 4$ | Shortest paths | 0.49 (0.06) | 0.41 (0.05) | 0.30 (0.03) |
|  | Noisy shortest paths | 0.44 (0.08) | 0.22 (0.06) | 0.33 (0.06) |
|  | Random walks | 0.64 (0.08) | 0.98 (0.02) | 0.51 (0.06) |

**Table 6:** Compression and distinction metrics for maps data where token acceptance is based on the top-$k$ decoding mechanism. We consider values from $k = 1$ to $k = 4$ because there are only 8 valid cardinal directions.

we see the tradeoff between precision and recall as $\epsilon$ changes. However, the conclusions are stable: every model has an incoherent world model across values of $\epsilon$. For example, while the random walks model has high compression and distinction precision for $\epsilon = 10^{-6}$, it has a very low distinction recall, of 0.11. Meanwhile, while the distinction recall is bumped up to 1.00 for $\epsilon = 0.10$, its compression precision falls to 0.16.

The metrics introduced in Section 2.4 depend on defining what it means for a model to accept or reject a sequence. In the main text we consider acceptance based on a threshold parameter, which corresponds to an $\epsilon$-sampling decoding mechanism [12]. Here, we consider two alternative forms of acceptance based on other decoding strategies: top-$k$ [8] and top-$p$ [13]. For acceptance based on top-$k$, a token is accepted if it's in the model's top-$k$ ranked tokens for a sequence and rejected otherwise. For top-$p$, a token is accepted if it's part of the the smallest set of highest-probability tokens whose cumulative probability is larger than $p$. Results are depicted in Table 6 and Table 7 and point to the same conclusion as the threshold-based metrics in Section 3; none of the models have recovered the world model, but the model trained on random walks performs best.

Our test also relies on Monte-Carlo sampling the model's Myhill-Nerode boundary. The number of samples affects only the precision test. A prefix pair fails the compression test whenever there's one suffix that the model accepts for one prefix and not for the other, so performance should worsen as the number of samples increases. Table 8 shows how the precision scores vary as a function of the number of samples for the shortest paths model. We use $M = 30$ Monte Carlo samples for our main reported metrics.

|  |  | Compression precision | Distinction precision | Distinction recall |
|---|---|---|---|---|
| $p = 0.90$ | Shortest paths | 0.05 (0.02) | 0.33 (0.05) | 0.17 (0.03) |
|  | Noisy shortest paths | 0.08 (0.04) | 0.34 (0.07) | 0.20 (0.05) |
|  | Random walks | 0.16 (0.06) | 0.96 (0.04) | 0.94 (0.03) |
| $p = 0.99$ | Shortest paths | 0.22 (0.05) | 0.39 (0.05) | 0.21 (0.03) |
|  | Noisy shortest paths | 0.03 (0.03) | 0.39 (0.07) | 0.24 (0.05) |
|  | Random walks | 0.54 (0.08) | 1.00 (0.00) | 1.00 (0.00) |
| $p = 0.999$ | Shortest paths | 0.31 (0.05) | 0.35 (0.05) | 0.22 (0.04) |
|  | Noisy shortest paths | 0.15 (0.06) | 0.46 (0.07) | 0.31 (0.05) |
|  | Random walks | 0.73 (0.07) | 1.00 (0.00) | 1.00 (0.00) |

**Table 7:** Compression and distinction metrics for maps data where token acceptance is based on the top-$p$ decoding mechanism.

| Number of samples | Compression precision | Distinction precision |
|---|---|---|
| 10 | 0.14 (0.04) | 0.35 (0.05) |
| 20 | 0.12 (0.04) | 0.35 (0.05) |
| 30 | 0.10 (0.03) | 0.30 (0.04) |

**Table 8:** Precision performance as a function of the number of Monte Carlo samples used to approximate a model's Myhill-Nerode boundary. Results are for the shortest paths map model.

# F   Rides data construction and training

Here we describe the rides dataset construction in more detail. Our empirical studies are based on a dataset of taxi rides in New York from 2014, originally released by the NYC Taxi & Limousine Commission. We use a subset of 15 million rides that took place between January and March 2014, made available by Murray [25]. We drop duplicate rides and subset the dataset to only include rides in Manhattan, resulting in 3,358,737 sequences. We use the OSMnx library [5] to represent New York as a weighted graph. We match pickups and dropoffs to the closest intersection, measured in terms of latitude/longitude. The graph consists of 4,580 nodes, 9,846 edges, and each node has a median of 2 valid intersections. We convert bearings to one of 8 cardinal directions. We remove short traversals with two node or less. We also remove sequences with more than 100 tokens.

**Shortest paths.**   The first approach creates traversals between two nodes by finding the shortest path between them. For each taxi ride in the dataset, we map the pickup latitude/longitude to the closest intersection and do the same for the dropoff location. We then perform Dijkstra's algorithm to find the shortest path weighted by distance. After filtering out duplicated traversals, we have a training set of 2,932,675 sequences and 120,400,201 tokens, along with a validation set of 1,000 sequences and 41,641 tokens.

**Noisy shortest paths.**   Shortest path traversals may not be reflective of real-world traversals due to differences in traffic patterns. Moreover, shortest path traversals between two nodes are deterministic, potentially limiting a model's ability to pick up a world model. Therefore, we construct a noisy version of shortest-path traversals. We do this by modifying the weighting function, $\tilde{W}(i, j) = W(i, j) + \epsilon_{ij}$, where $\epsilon_{ij} \sim \text{Gamma}(W(i, j), 1)$; we can interpret this as artificially adding traffic to each edge that scales with the original length. We resample 50 different weighting functions. After filtering out duplicated traversals, we have a training set with 30,599,312 sequences and 1,677,587,216 tokens while our validation set consists of 1,000 sequences and 54,539 tokens

**Random walks.**   In the last setting, we sample random traversals rather than approximating shortest paths. We construct each sequence by sampling a node uniformly at random, sampling a sequence length uniformly between 3 and 100, and constructing traversals by sampling random edges uniformly at random for the prespecified sequence length. We create a training set of 90,646,864 sequences and 4,735,591,368 tokens, along with a validation set of 1,000 sequences and 52,360 tokens.

We use the GPT-2 architecture [29] to train models on all datasets. We use the GPT-2 small architecture for the shortest paths model and the GPT-2 extra-large architecture for the noisy shortest paths and

random walks models. For the shortest paths models, we train until we overfit and use the best validation checkpoint. For the two larger datasets, we train for a fixed number of epochs and use the last validation checkpoint; we use 5 epochs for the noisy shortest paths dataset and 1 epoch for the random walks dataset. We train all models on 8 A100 GPUs, using a batch size of 6 sequences per GPU. Training time ranges from about 12 hours for the shortest paths model to 48 hours for the random walks model.

## G    Additional maps

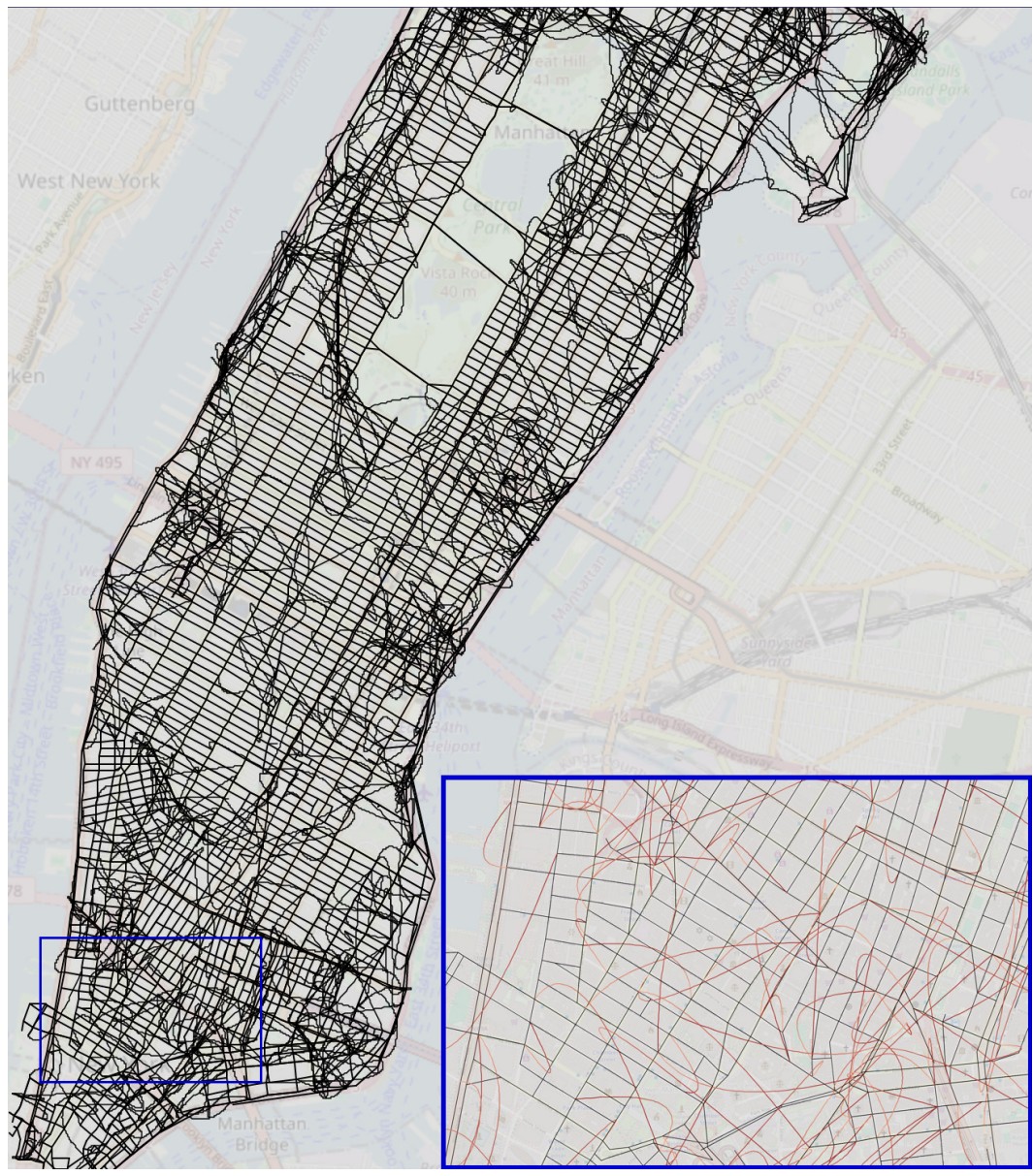

**Figure 9:** Reconstructed map from transformer trained on **shortest paths**. In the zoomed-in images, edges belonging to the true graph are black and false edges added by the reconstruction algorithm are red with a darkening gradient indicating the directionality of the edge. Interactive map available at https://manhattan-reconstruction-shortest.netlify.app/.

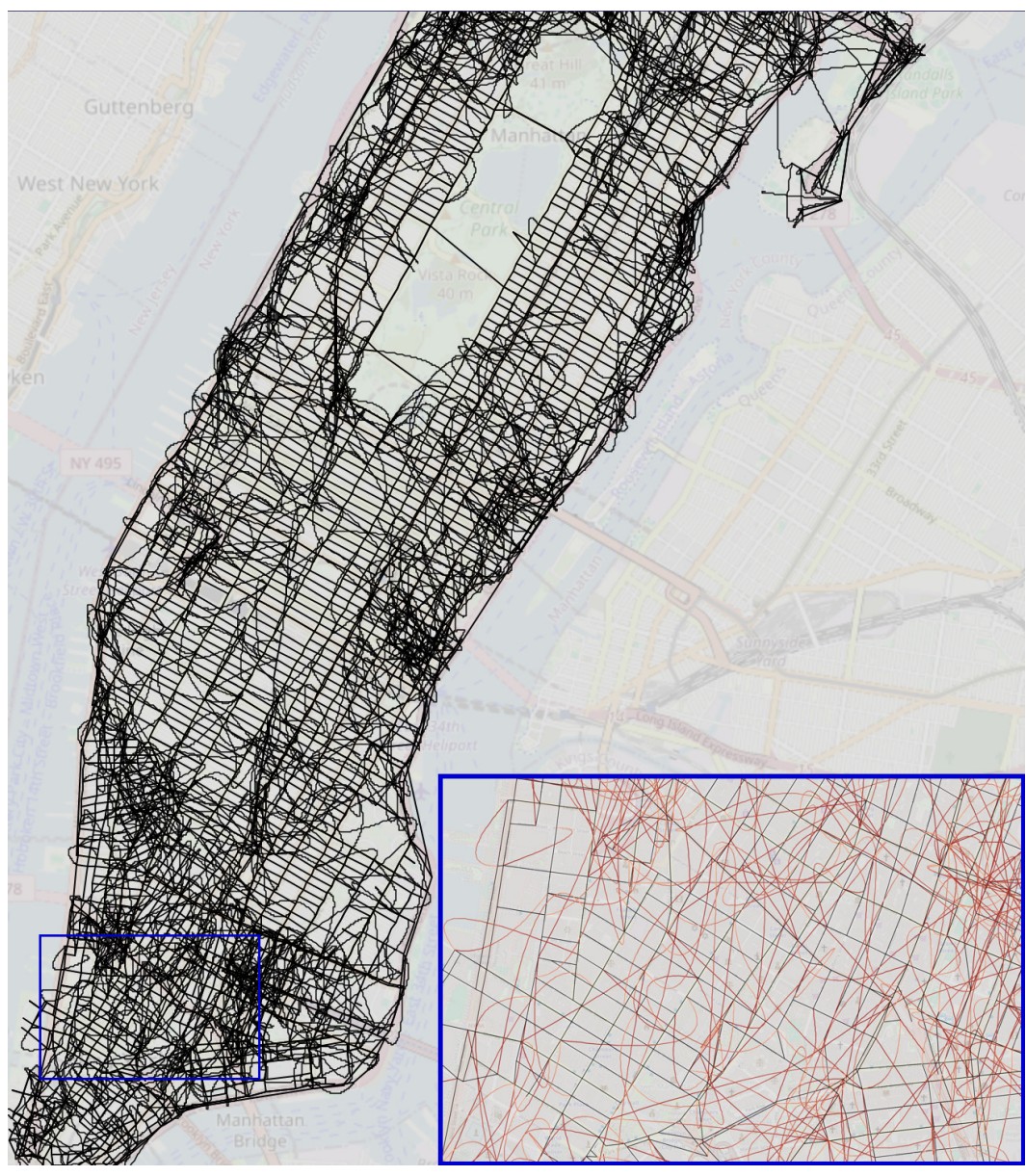

**Figure 10:** Reconstructed map from transformer trained on **noisy shortest paths**. Edges exit nodes in their specified cardinal direction. Interactive map available at `https://manhattan-reconstruction-noisy.netlify.app/`.

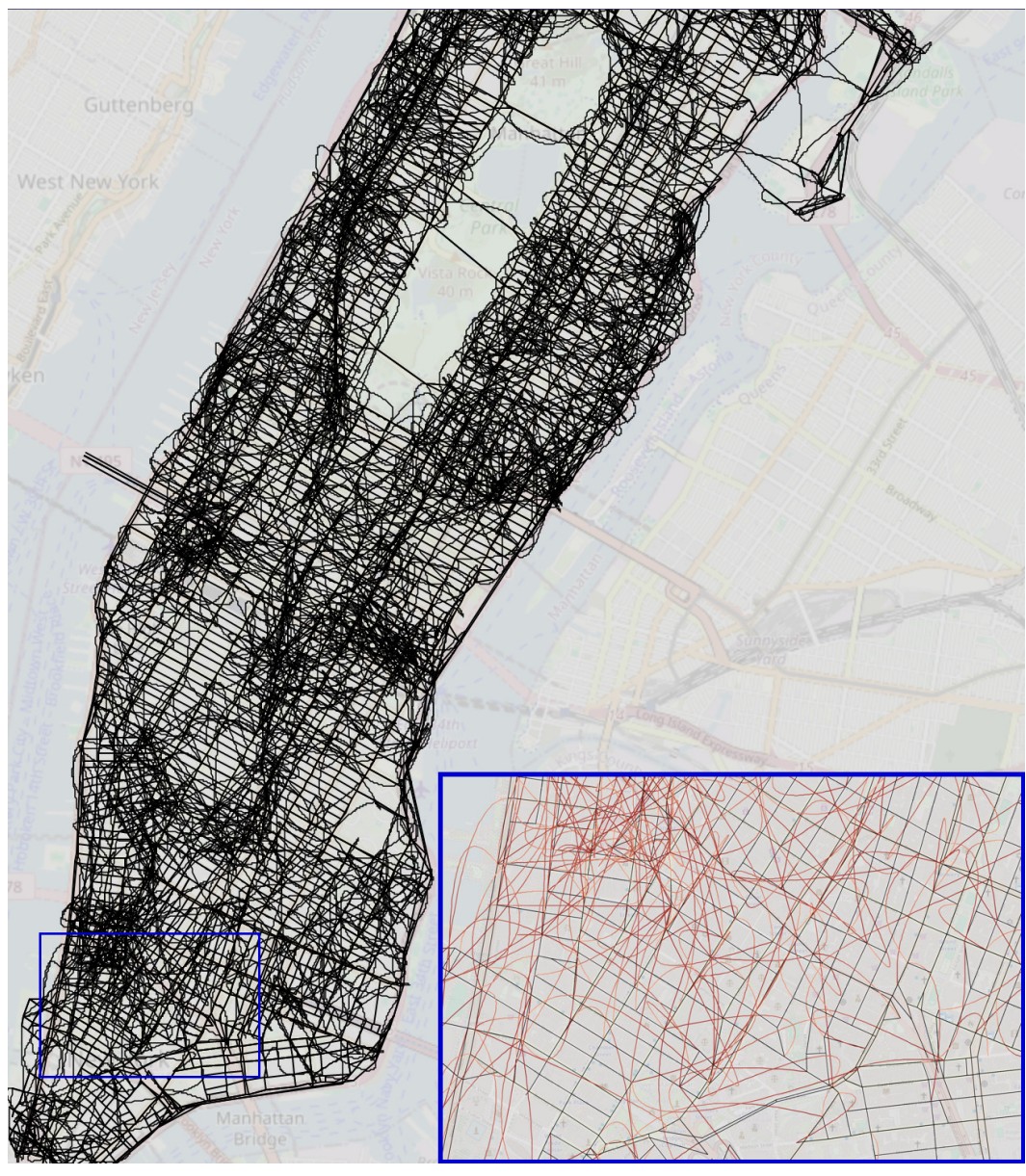

**Figure 11:** Reconstructed map from transformer trained on **random walks**. Interactive map available at https://manhattan-reconstruction-random.netlify.app/.

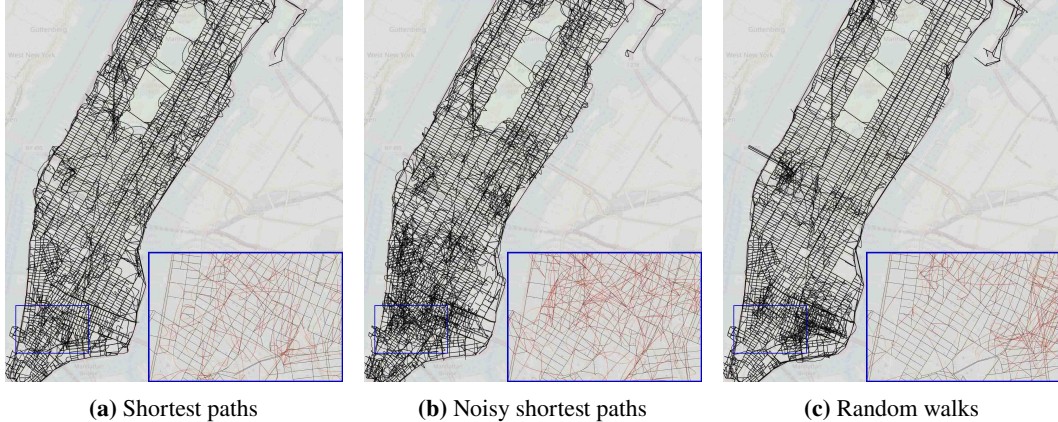

**(a)** Shortest paths      **(b)** Noisy shortest paths      **(c)** Random walks

**Figure 12:** Reconstructed maps from transformers trained on shortest paths, noisy shortest paths, and random walks constructed using (origin, destination) pairs sampled from the same distribution used for training. While trajectories are valid more than 95% of the time, the errors reveal incoherence. Reconstruction uses 50k sequences with maximum degree 4 and maximum edge length 1/2 mile.

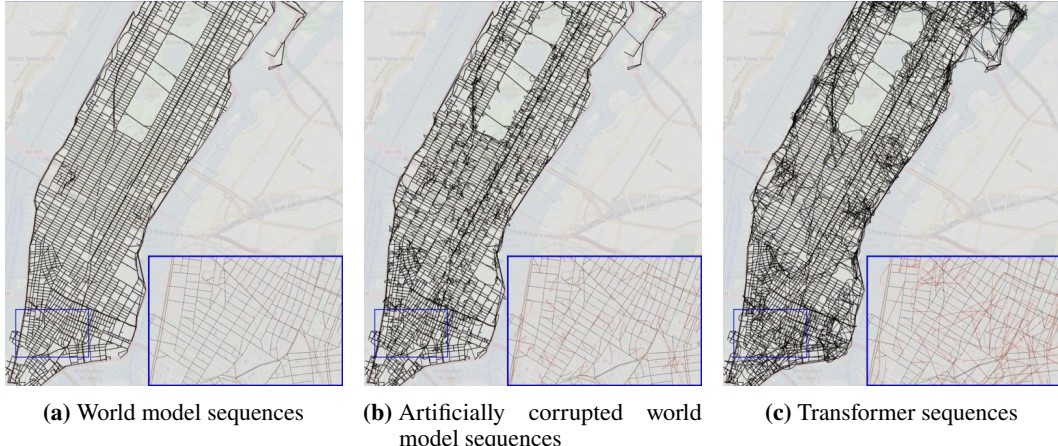

**(a)** World model sequences      **(b)** Artificially corrupted world model sequences      **(c)** Transformer sequences

**Figure 13:** Reconstructed maps of Manhattan from sequences produced by the generative model trained on **shortest paths with maximum degree** 4 **and the maximum edge length** 1/2 **mile**. The reconstruction process failed on 1513/6400 sequences.

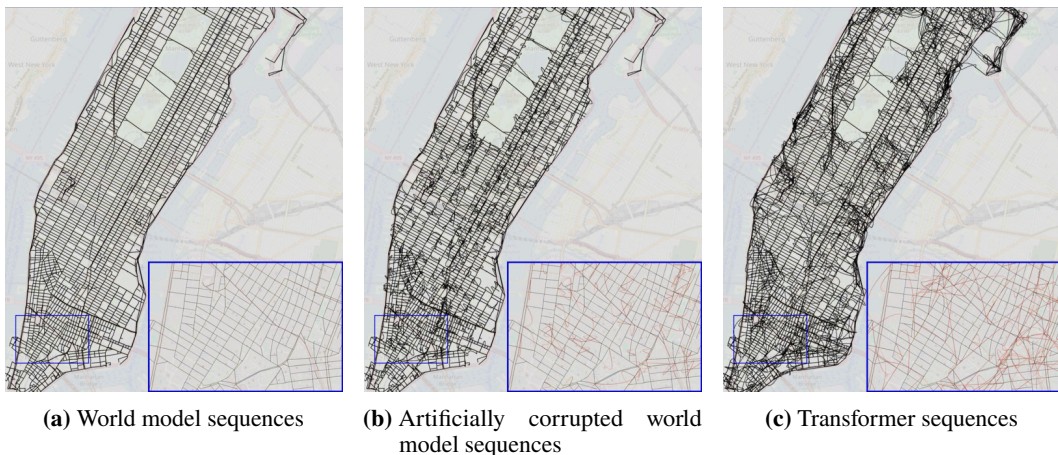

**(a)** World model sequences      **(b)** Artificially corrupted world model sequences      **(c)** Transformer sequences

**Figure 14:** Reconstructed maps of Manhattan from sequences produced by the generative model trained on **shortest paths with maximum degree** 8 **and the maximum edge length** 1/2 **mile**. The reconstruction process failed on 1435/6400 sequences.

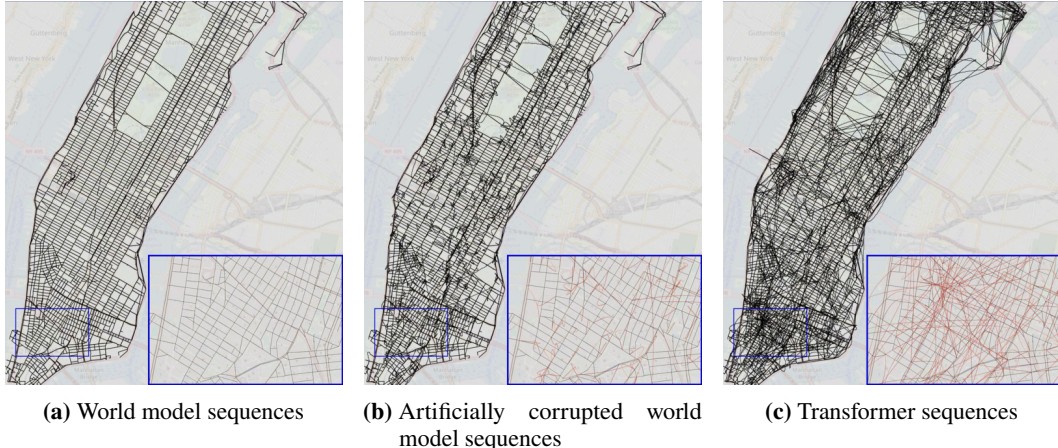

**(a)** World model sequences

**(b)** Artificially corrupted world model sequences

**(c)** Transformer sequences

**Figure 15:** Reconstructed maps of Manhattan from sequences produced by the generative model trained on **shortest paths with maximum degree** 4 **and the maximum edge length** 1 **mile**. The reconstruction process failed on 1334/6400 sequences.

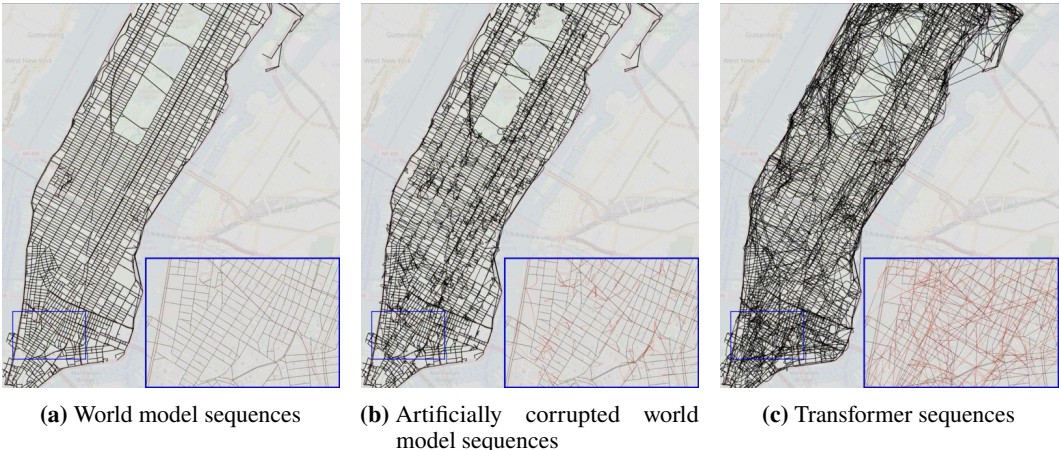

**(a)** World model sequences

**(b)** Artificially corrupted world model sequences

**(c)** Transformer sequences

**Figure 16:** Reconstructed maps of Manhattan from sequences produced by the generative model trained on **shortest paths with maximum degree** 8 **and the maximum edge length** 1 **mile**. The reconstruction process failed on 1174/6400 sequences.

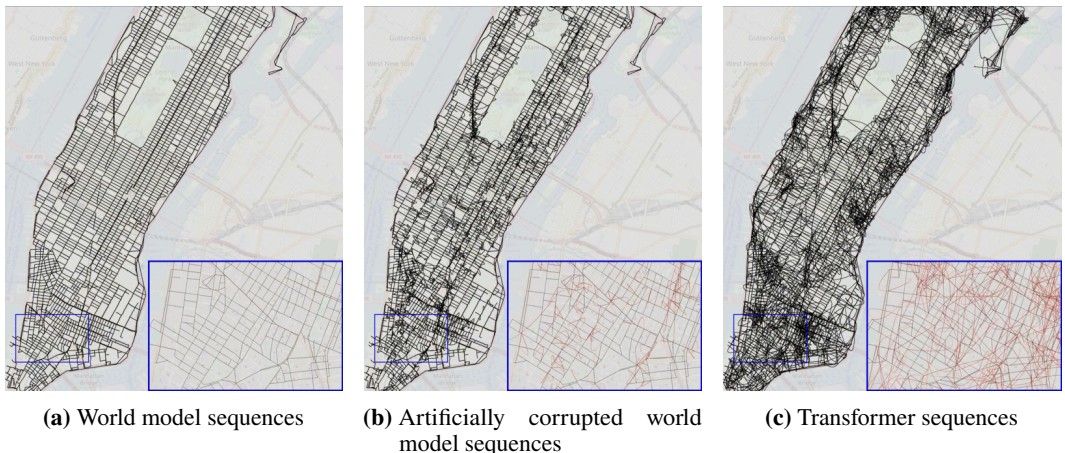

**(a)** World model sequences

**(b)** Artificially corrupted world model sequences

**(c)** Transformer sequences

**Figure 17:** Reconstructed maps of Manhattan from sequences produced by the generative model trained on **noisy shortest paths with maximum degree** 4 **and the maximum edge length** 1/2 **mile**. The reconstruction process failed on 2213/6400 sequences.

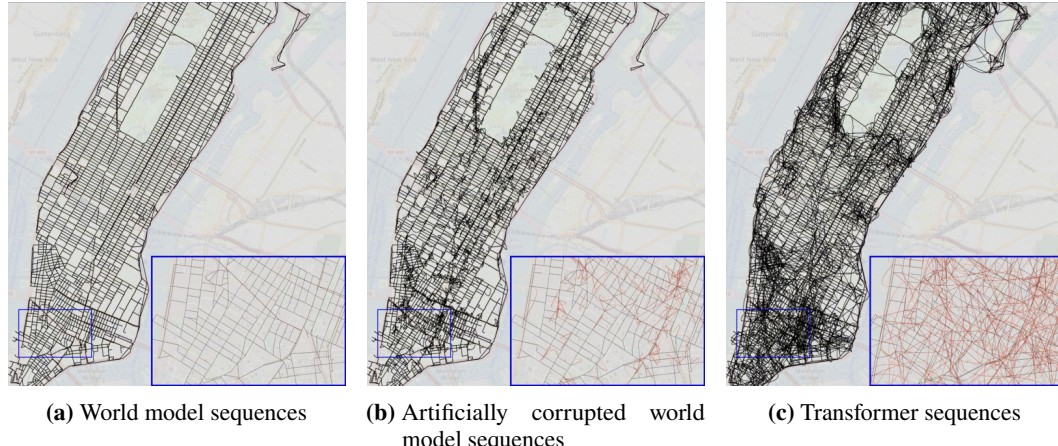

**(a)** World model sequences   **(b)** Artificially corrupted world model sequences   **(c)** Transformer sequences

**Figure 18:** Reconstructed maps of Manhattan from sequences produced by the generative model trained on **noisy shortest paths with maximum degree** 8 **and the maximum edge length** 1/2 **mile**. The reconstruction process failed on 1869/6400 sequences.

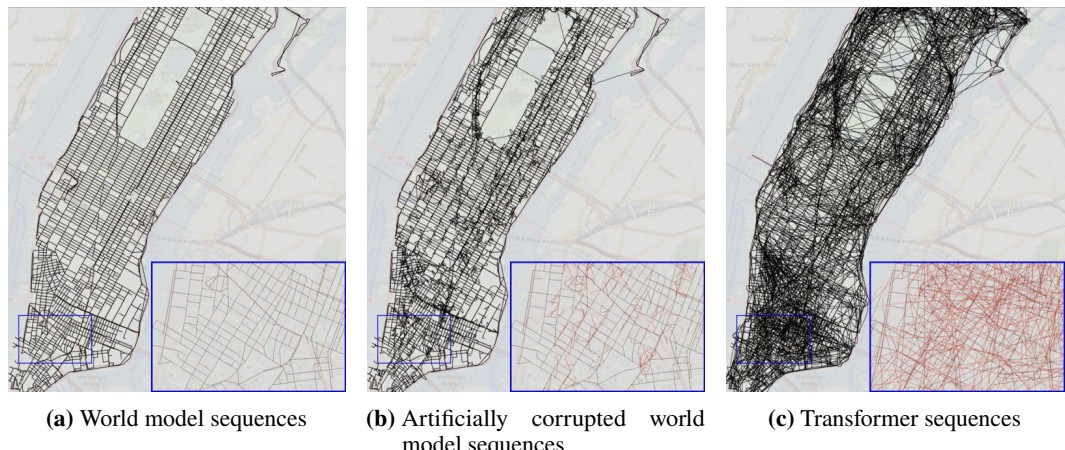

**(a)** World model sequences   **(b)** Artificially corrupted world model sequences   **(c)** Transformer sequences

**Figure 19:** Reconstructed maps of Manhattan from sequences produced by the generative model trained on **noisy shortest paths with maximum degree** 4 **and the maximum edge length** 1 **mile**. The reconstruction process failed on 1935/6400 sequences.

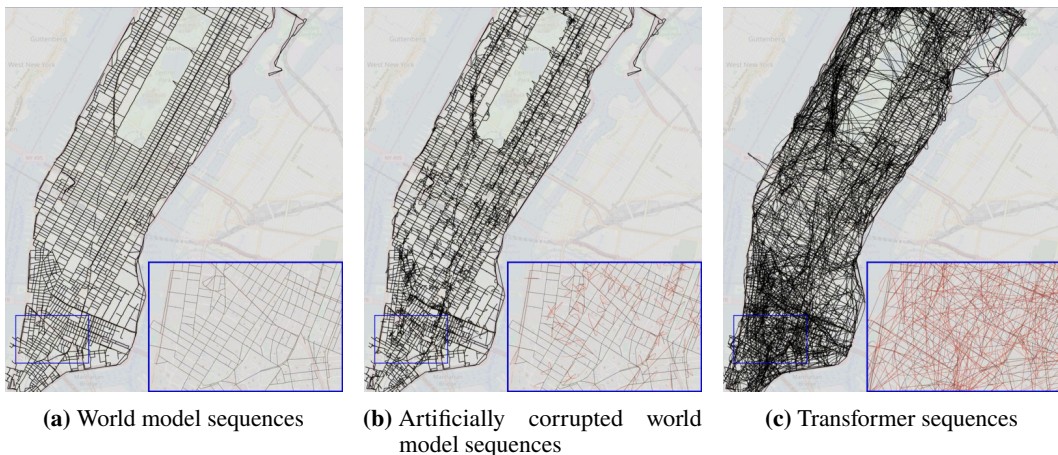

**(a)** World model sequences   **(b)** Artificially corrupted world model sequences   **(c)** Transformer sequences

**Figure 20:** Reconstructed maps of Manhattan from sequences produced by the generative model trained on **noisy shortest paths with maximum degree** 8 **and the maximum edge length** 1 **mile**. The reconstruction process failed on 1702/6400 sequences.

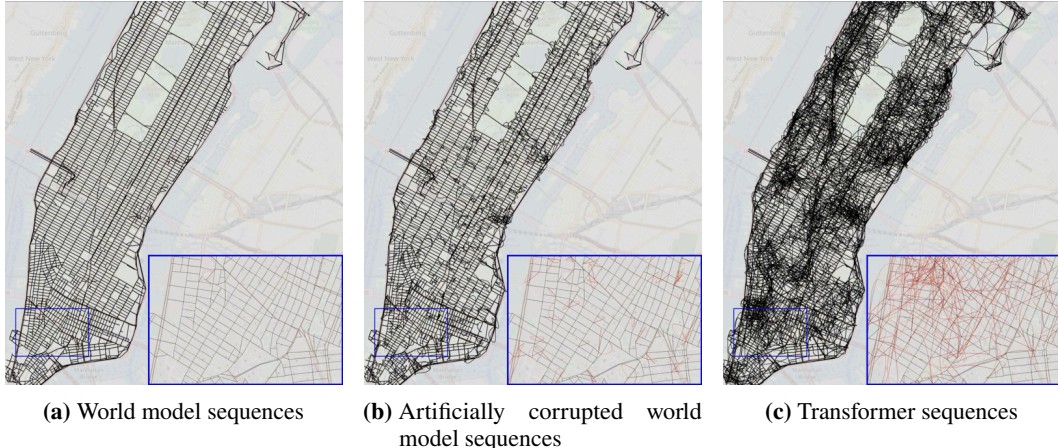

**(a)** World model sequences

**(b)** Artificially corrupted world model sequences

**(c)** Transformer sequences

**Figure 21: (Copy of Figure 3)** Reconstructed maps of Manhattan from sequences produced by the generative model trained on **random walks with maximum degree** 4 **and the maximum edge length** 1/2 **mile**. The reconstruction process failed on 1987/6400 sequences.

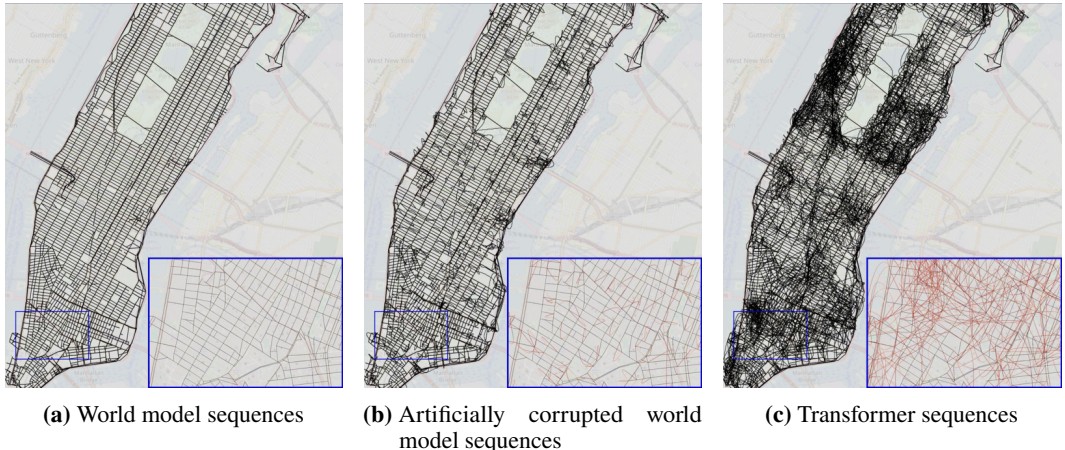

**(a)** World model sequences

**(b)** Artificially corrupted world model sequences

**(c)** Transformer sequences

**Figure 22:** Reconstructed maps of Manhattan from sequences produced by the generative model trained on **random walks with maximum degree** 8 **and the maximum edge length** 1/2 **mile**. The reconstruction process failed on 1491/6400 sequences.

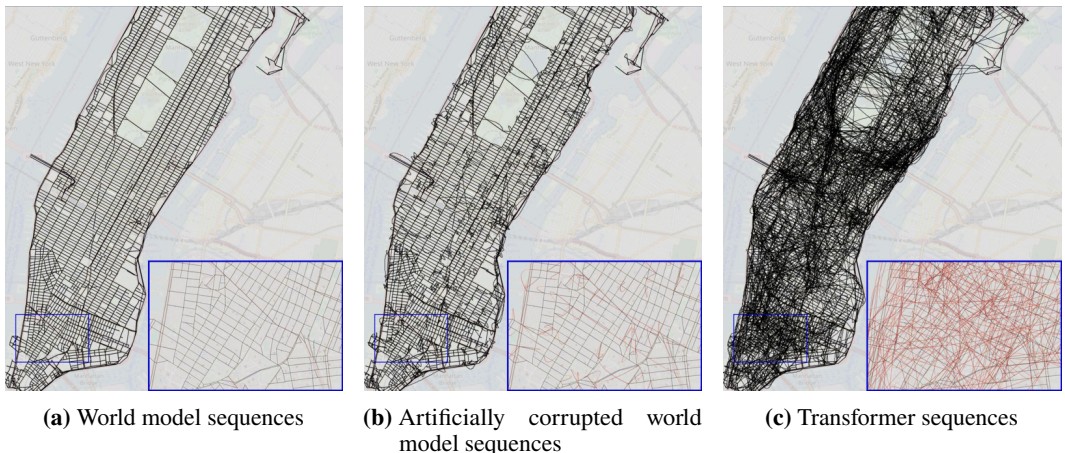

**(a)** World model sequences

**(b)** Artificially corrupted world model sequences

**(c)** Transformer sequences

**Figure 23:** Reconstructed maps of Manhattan from sequences produced by the generative model trained on **random walks with maximum degree** 4 **and the maximum edge length** 1 **mile**. The reconstruction process failed on 1905/6400 sequences.

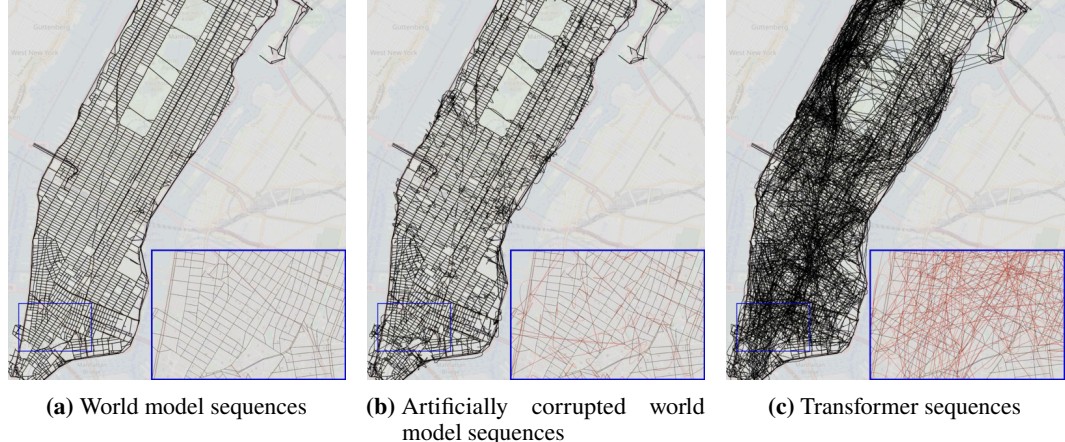

**(a)** World model sequences  **(b)** Artificially corrupted world model sequences  **(c)** Transformer sequences

**Figure 24:** Reconstructed maps of Manhattan from sequences produced by the generative model trained on **random walks with maximum degree** 8 **and the maximum edge length** 1 **mile**. The reconstruction process failed on 1323/6400 sequences.

