# OpenReview forum: "Evaluating the World Model Implicit in a Generative Model"
_NeurIPS.cc/2024/Conference — NeurIPS 2024 spotlight_

### Official Review · Reviewer_S5LR · 2024-06-16

**Soundness:** 3
**Presentation:** 3
**Contribution:** 2
**Rating:** 4
**Confidence:** 4

**Summary:**

This paper aims to develop new metrics for assessing a model’s ability to recover a world model. The key idea is to test coherence with respect to the world, guided by the Myhill-Nerode theorem for deterministic finite automata (DFA). Specifically, if the true world model is a DFA, the learned world model should meet two requirements: (1) sequences leading to the same DFA state must have the same continuations, and (2) sequences leading to distinct DFA states should have distinct continuations. Based on these principles, the authors proposed three new metrics: compression precision, distinction precision, and distinction recall.

To test these metrics, they trained a GPT model on New York City taxi rides, hoping the GPT would learn a map of NYC. They found that the trained transformer model performed nearly perfectly using classical methods for testing world models, such as next-token prediction and state probe. However, under their three proposed metrics, the trained model appeared not to learn the world model at all, suggesting that these new metrics might be a valuable alternative for evaluation.

**Strengths:**

This work is well-motivated and well-written. The proposed metrics are interesting, and the authors have supplemented the study with detailed ablation studies on the training data, which is commendable.

**Weaknesses:**

1. Generalizability of the proposed metrics

The main concern I have with the proposed metrics is their generalizability. The metrics are strictly applicable when the true world model is a DFA. While this might be suitable for the New York City map and the Othello game, it is not directly applicable to Logic Puzzles. The issue is that Logic Puzzles are prompted using natural language (see Fig. 22), and natural language texts cannot be accurately modeled as DFAs. Although one could argue that the same state is reached via different prompts at the latent concept level, language models do not operate directly on this latent concept space. Instead, they work on the token level, which cannot be represented as a DFA.

This limitation significantly restricts the scenarios in which the proposed metrics can be applied. In contrast, the two existing metrics (next-token prediction and state probe) are applicable regardless of the true world model. This fundamental limitation could impact the overall significance and applicability of the study.

2. Inter-metric consistency problem

Despite the limitations of the proposed metrics, they would still be valuable if they performed well under the DFA assumption in indicating world model recovery. However, there is a lack of inter-metric consistency. In Table 1 (Random Walks row), almost all metrics, except for the proposed Compression Precision metric, indicate that the model perfectly captures the world model. This inconsistency raises questions: What is the correct conclusion when two out of three metrics suggest the presence of a world model, while the other does not? How can these metrics be relied upon if there are substantial internal inconsistencies based on the statistics motivated by the Myhill-Nerode theorem?

Resolving this issue in practice may be challenging. It appears that the three proposed metrics have low false negatives (i.e., they perform well when the model does not learn the true world model, similar to existing metrics). However, they seem to have high false positives (i.e., the statistics struggle to detect if a model has actually learned the world model). Resolving this issue may be difficult because it likely requires substantial knowledge of the true world model to develop the appropriate statistical corrections for sampling two sequences that reach the same DFA state.

**Questions:**

What was the context length for your GPT model? This is crucial, as a context length that is too short for the type of data presented to the model will negatively impact its performance.

**Limitations:**

The main text lacks an explicit section on Limitations and Future Research. However, the authors acknowledged in the Conclusion that their primary limitation was the focus on deterministic finite automata.

Minor comments:

Line 76: “our evaluation metrics are based sequences” —> “our evaluation metrics are based [on] sequences”

---

> ### Author Rebuttal · Authors · 2024-08-07
>
> Thank you for your insightful review. We're glad you found our proposed metrics interesting and ablation studies compelling. We appreciate your comments on the clarity of our paper.
>
> > _Inter-metric consistency problem... What is the correct conclusion when two out of three metrics suggest the presence of a world model, while the other does not?_
>
> You raise a great question. If a model has the true world model, all metrics will score 100%. Conversely, if any metric is less than 100%, it won't have a perfect world model. As you note, once the metrics aren't perfect, some can be worse than others. This is similar to supervised learning: AUC, accuracy, F1, etc. are all the same when we have a perfect classifier. It's only when we have an imperfect model that these metrics reach different conclusions. This is why having multiple metrics is important: they tell you where a model is failing.
>
> For example, Table 1 shows that while the random walks model is able to differentiate between states, this comes at the expense of failing to capture that the same state can be reached by different sequences. There is a tradeoff between metrics: it's easy to ace compression (by saying every sequence has the same continuations) but then distinction suffers.
>
> > _It appears that the three proposed metrics have low false negatives [and]... high false positives_
>
> This fantastic comment helped us clarify the discernment properties of our proposed metrics. A model with the true world model will not fail when detours are introduced to sequences. In our original submission, we used detours to validate our metrics on the navigation exercise; detour performance correlated with our metrics (Table 2). We perform the same exercise below to validate our metrics' discernment of Othello models:
>
> Existing metrics imply that both OthelloGPT models from [1] ("Championship" and "Synthetic") are close to having the true world model. Our metrics reveal a more complex picture: while Synthetic recovers the true world model, Championship fails by our metrics.
>
> Crucially, we can validate this discernment with an additional "detour" exercise: with probability p, we replace a model's top predicted move with another valid move and assess whether it completes the game validly. While the Synthetic model produces near-perfect games regardless of detours, the Championship model fails immediately. A model that recovers the true world model will succeed regardless of detours. There is a clear distinction between Championship and Synthetic models, but this is only captured by our metrics; existing metrics would lead us to conclude that both have world models.
>
> **Random detours**
> |Model|0%|1%|10%|25%|50%|
> |-|-|-|-|-|-|
> |Championship|1.00 (0.00)|0.66 (0.05)|0.05 (0.02)|0.01 (0.01)|0.01 (0.01)|
> |Synthetic|1.00 (0.00)|0.99 (0.01)|0.97 (0.02)|0.97 (0.02)|0.99 (0.01)|
>
> **Adversarial detours**
> |Model|0%|1%|10%|25%|50%|
> |-|-|-|-|-|-|
> |Championship|1.00 (0.00)|0.70 (0.05)|0.01 (0.01)|0.01 (0.01)|0.00 (0.00)|
> |Synthetic|1.00 (0.00)|0.98 (0.01)|0.99 (0.01)|0.96 (0.02)|0.97 (0.02)|
>
> > _The main concern I have with the proposed metrics is their generalizability. The metrics are strictly applicable when the true world model is a DFA. While this might be suitable for the New York City map and the Othello game, it is not directly applicable to Logic Puzzles_
>
> You bring up an important point to clarify on logic puzzles. Constant-sized logic puzzles are canonical examples of DFAs [2]. Because each puzzle corresponds to a true state, we can assess the different ways state is reflected by a model; this is why [3] use logic puzzles to probe a model's representations for state, and it's why we can study token-level outcomes to assess whether a model's behavior is consistent with state. While it may seem like the model is performing badly on our test because we're translating logic into natural language that can confuse LLMs, the input space of possible sequences is relatively simple (21 possible statements). LLMs perform poorly on our metrics despite this simplicity. We agree that adding richer language would create more realism; still if a model performs poorly with the simple language, it's informative for their understanding of more complex problems.
>
> Generally, we focus on DFAs because they're common in real-world phenomena: we focus on game-playing, logic, and navigation, and they also arise in search engines, control systems, and genetics [3]. Papers that study world model recovery have also focused on DFAs, even if they don't explicitly mention it [4, 5, 6]. DFAs are also important to study as testbed problems: What can we say about an LLM’s world model if it can’t recover a map?
>
> Finally, the requirements for our evaluation metrics fall in line with existing metrics in this literature: sampling from a model, seeing its predictions, and comparing them to the set of allowed predictions. These are the same requirements as the next-token test, and are similar to the probe test, which additionally requires access to a model's internal representations.
>
> > _What was the context length for your GPT model?_
>
> The maximum navigation length during training was 100 tokens so we used a context length of 100. Across all tests, we were careful to only evaluate models using the context length they were trained on, e.g. when we sample sequences for navigating from A to B we made sure there was a valid route with at most 100 moves (see lines 299-300 and 508-510). We did the same thing for Othello (which was trained on up to 60 tokens since games have <=60 moves).
>
> Thanks again for your review. If we've addressed your comments, we hope you'd consider raising your score.
>
> [1] https://arxiv.org/abs/2210.13382
> [2] https://link.springer.com/chapter/10.1007/978-3-642-59126-6_7
> [3] https://www.cs.ucdavis.edu/~rogaway/classes/120/spring13/eric-dfa.pdf
> [4] https://arxiv.org/abs/2106.00737
> [5] https://arxiv.org/abs/2210.13382
> [6] https://arxiv.org/abs/2102.13249

---

> > ### Comment · Reviewer_S5LR · 2024-08-12
> >
> > Thank you for the rebuttal. While some of my concerns were addressed, I remain concerned about the applicability of the developed metric. It appears that this metric is effective primarily when the true world model has been fully learned. In cases where the world model is imperfectly learned, the metric may not be reliable, as it does not provide directly comparable scores among imperfect models. Although achieving 100% on all metrics indicates that the true DFA has been learned, scores below 100% only confirm that the model is imperfect, without offering a clear measure of the degree of imperfection.

---

> > > ### Author Response · Authors · 2024-08-12
> > >
> > > Thank you for engaging with our paper. We're glad our rebuttal addressed some of your concerns. Here we attempt to answer your remaining questions:
> > > - We want to clarify a potential confusion: while it's correct that any metric less than 100% implies there is no world model, the **metrics can also be directly compared to one another.** This is again similar to supervised learning: while there's a tradeoff between false positive rate and false negative rate, if one model always has a better false positive rate and false negative rate, it is doing better than another model, even if it's imperfect. Our metrics are not arbitrary scores; they are grounded in a theoretically-based formulation of what it means to learn the structure of a world model. We also validate this empirically: for each model (across maps and Othello), a model's ranking on our metric is exactly its ranking on the detours exercise (for all three metrics and both kinds of detour exercises).
> > > - Our results would still be interesting even if our metrics only captured whether there is a world model or not. This is because **prior metrics would lead us to conclude that all models we test across maps and Othello do have world models.** Our metrics are capturing something new, and this is not only motivated theoretically but also validated empirically on 1) map reconstruction and 2) detour exercises on Othello and maps.

---

### Official Review · Reviewer_RjHx · 2024-07-10

**Soundness:** 2
**Presentation:** 3
**Contribution:** 2
**Rating:** 3
**Confidence:** 5

**Summary:**

This paper proposes a new metric to assess the implicit world model of generative models, such as neural LMs. Inspired by the Myhill-Nerode theorem, this metric evaluates whether a model can determine if pairs of sequences are equivalent in terms of their underlying state. The author presents two specific metrics: sequence compression (SC) and sequence distinction (SD). For a pair of sequences (e.g., natural text, destination trajectories, or game scripts) that correspond to the same state, SC measures whether neural LMs accept the same set of continuations. Conversely, SD assesses whether neural LMs can accurately distinguish between sets of continuations that are uniquely permissible for one sequence but not for the other.

The paper assumes that the model under test is a Deterministic Finite Automaton (DFA), and hence the model can directly determine whether a sequence is accepted. However, since neural LMs lack this capability, the author suggests using a token-level threshold as a proxy: $\forall t  P_\theta(x_t|x_{1:t-1})> \epsilon$.

Experiments conducted across three datasets with various neural LMs (small-scale Transformers & LLMs) indicate that these models score significantly lower on SC&SD compared to existing metrics. Based on these findings, the authors claims that SC&SD are more faithful.

**Strengths:**

1. This work proposes a new metric that measures the coherence of the implicit world model of neural LMs, which is novel perspective in this field.
2. The paper is overall well-written and easy to follow.
3. The authors conduct extensive evaluation on a wide range of base models (LLMs & Transformers) and three datasets, and the results are consistent.

**Weaknesses:**

1. The authors inaccurately summarize existing work and consequently address a non-existent flaw.

>... Toshniwal et al. and Li et al. assess world models by measuring the percent of top model predictions that are valid in the underlying DFA. (L112-113)

This is not true. Li et al. [1] do not solely rely on the validity of the top model predictions to assess the implicit world model. Instead, they directly probe the internal world state from the model, which has been the common practice of existing work ([2], [3]).

2. There are two fundamental flaws of the proposed metric
     * accountability issue: Poor performance according to the metric could stem from an bad transformation of actions from the world state, rather than a bad implicit world model.
     * It's challenging to ascertain whether a neural LM "accepts" a sequence, which makes it hard for the theoretical guarantees of SC and SD hold in practice. The proposed metric relies heavily on the deterministic nature of the DFA, which directly indicates sentence acceptance. In contrast, neural LMs model distributions over tokens or sentences without a clear mechanism for determining sentence acceptance. The author proposes using a token-level probability threshold as a workaround, yet this approach has several flaws:
       * A top-$k$ predicted sequence may include tokens with low likelihood.
       * Conversely, a sequence satisfying the criteria might not appear in the top-$k$ predictions.
       * Additional confounding variables can affect the metric's value. For example, LMs with higher entropy typically have a larger set of "accepted" sequences. This entropy can be influenced by the hyperparameters of decoding algorithms, training, and fine-tuning methods of the LMs, etc.

3. Lack of empirical evidence that proves the faithfulness of the proposed metric.
     * Considering the loose approximation of the DFA-style acceptance, I expect the author to provide empirical justification of the proposed metric, which is notably missing in the paper. The only evidence presented is the poor performance of existing LLMs on this metric. However, this is insufficient since the lower performance on the metric could be simply because the methods are unfairly penalized.
    * Also, there is a concerning indication of the metric's questionable faithfulness: the conclusions drawn from the Othello experiments in this paper contradict those of Li et al. [1]. Li et al. convincingly probes Transformer's internal representation of the board state and demonstrates that it can causally influence the Transformers' predictions.

[1] Li, Kenneth, et al. "Emergent world representations: Exploring a sequence model trained on a synthetic task." ICLR 2022.

[2] Yun, Tian, et al. "Emergence of Abstract State Representations in Embodied Sequence Modeling." Proceedings of the 2023 Conference on Empirical Methods in Natural Language Processing. 2023.

[3] Karvonen, Adam. "Emergent world models and latent variable estimation in chess-playing language models." arXiv preprint arXiv:2403.15498 (2024).

**Questions:**

1. Did you try other acceptance criteria that go beyonds manipulation of threshold value, e.g. sequence-level probability/rank?

**Limitations:**

The author doesn't address the limitations discussed above in the paper.

---

> ### Author Rebuttal · Authors · 2024-08-07
>
> Thank you for your review. Your review makes several helpful points that will improve our paper. However our rebuttal clarifies a couple of important points of your review: one involving an incorrect statement of what is in our paper (we do describe and empirically test probes), and another is a clarification of the value of having multiple metrics. We'll make these more clear in our revision.
>
> > _The authors inaccurately summarize existing work... Li et al. [1] do not solely rely on the validity of the top model predictions_
>
> Your review states that we ignore the probe test of Li et. al [1]. We agree probes are important and this paper would be incomplete without them. However, we not only describe the state probe in the Li et al. paper and the larger literature in the related work section (lines 74-78) but also perform probing experiments for our maps setting (Table 1 and lines 240-242). We agree that the sentence you point out could do more to foreshadow these results/discussion, so we'll update it in the revision and cite the two additional papers you mention.
>
> > _The conclusions drawn from the Othello experiments in this paper contradict those of Li et al._
>
> Our tests measure different outcomes than Li et al [1]. If a model's world model is perfect, all metrics will score 100%. Conversely, if any metric is less than 100%, it won't have a world model. This is similar to supervised learning: AUC, accuracy, F1, etc. are all the same for perfect classifiers. But having multiple metrics tells us _how_ an imperfect model is failing.
>
> We validated our metrics in the original submission by showing 1) they capture behavior other metrics don't and 2) they correlate with detour performance for the taxi exercise (Table 2). We now include additional validation for Othello using detours.
>
> Results from [1] imply that both OthelloGPT models ("Championship" and "Synthetic") are close to having true world models. Our metrics reveal a more complex picture: while Synthetic recovers the true world model, Championship fails by our metrics.
>
> Crucially, we can validate this differentiation. We do this with an additional "detour" exercise: with probability p, we replace a model's top predicted move with another valid move and assess whether it completes the game validly. While the Synthetic model produces near-perfect games regardless of detours, the Championship model fails immediately. There is a clear distinction between Championship and Synthetic models, but this is only captured by our metrics; existing metrics would lead us to conclude that both have world models.
>
> **Random detours**
> |Model|0%|1%|10%|25%|50%|
> |-|-|-|-|-|-|
> |Championship|1.00|0.66|0.05|0.01|0.01|
> |Synthetic|1.00|0.99|0.97|0.97|0.99|
>
> See the PDF for similar results on adversarial detours.
>
> > _Poor performance according to the metric could stem from an bad transformation of actions from the world state, rather than a bad implicit world model._
>
> Our metrics test a model's outputs while probes test a model's internal state. Both metrics are important because they measure different things: our metrics find inconsistencies not revealed by probes (see Tables 1, 2, 6, and the Othello detours above).
>
> Further, the reliability of probes is far from settled. Ongoing debate [2] includes issues like proper baselines [3, 4], classifier complexity [5, 6], and faithfulness [7, 8]. Even for OthelloGPT, probes are sensitive to specification: linear and nonlinear probes have very different error rates [1], and different encodings reach different conclusions [9]. Moreover, should probe labels reflect each tile individually (as [1] considers) or the full board (all 64 tiles)? The championship Othello probe has 91% accuracy for each tile individually [1], but accuracy falls to 0.2% when the label is the full board. We think probes are valuable tools, but they don't yet provide conclusive results given these open questions.
>
> > _Neural LMs model distributions over tokens or sentences without a clear mechanism for determining sentence acceptance._
>
> We agree that focusing on a single threshold alone provides an incomplete picture. This is why we include ablations across multiple thresholds (Table 3). All thresholds result in incorrect world models.
>
> Common decoding strategies _do_ provide mechanisms for determining sequence acceptance, like top-k, top-p, and threshold-based sampling (i.e. epsilon sampling) [10, 11, 12]. While our paper focuses on epsilon sampling [12], different choices can easily be made within our framework. Below we include additional results for top-p and top-k sampling (more ablations in the PDF), which are very similar to the original results. The paper will be stronger thanks to your suggestion to add these.
>
> **Top-p (p=0.99)**
> |Model|Compression precision|Distinction precision|Distinction recall|
> |-|-|-|-|
> |Shortest paths|0.22|0.39|0.21|
> |Noisy shortest paths|0.03|0.39|0.24|
> |Random walks|0.54|1.00|1.00|
>
> **Top-k (k=2)**
> |Model|Compression precision|Distinction precision|Distinction recall|
> |-|-|-|-|
> |Shortest paths|0.21|0.32|0.17|
> |Noisy shortest paths|0.07|0.31|0.23|
> |Random walks|0.21|0.93|0.73|
>
> Thanks again for your review. We think these edits will make the paper stronger. If we have addressed your comments, we hope you'd consider raising your score.
>
> [1] https://arxiv.org/abs/2210.13382
> [2] https://arxiv.org/abs/2102.12452
> [3] https://arxiv.org/abs/1805.01070
> [4] https://arxiv.org/abs/2004.03061
> [5] https://arxiv.org/abs/1610.01644
> [6] https://arxiv.org/abs/2104.03514
> [7] https://arxiv.org/abs/1812.08951
> [8] https://arxiv.org/abs/2004.14975
> [9] https://arxiv.org/abs/2309.00941
> [10] https://arxiv.org/abs/1904.09751
> [11] https://arxiv.org/abs/1805.04833
> [12] https://arxiv.org/abs/2210.15191

---

> ### Comment · Reviewer_RjHx · 2024-08-12
>
> Thank you for your rebuttal and the updated results. My responses are below.
>
>
> ## Summary of existing work
>
> ```
>  Your review states that we ignore the probe test of Li et. al [1].
> ```
>
> No, it doesn't.
>
> My criticism that *"The authors inaccurately summarize existing work and consequently address a non-existent flaw"* is directly related to the specific statement I referenced from your manuscript: *"Toshniwal et al. and Li et al. assess world models by measuring…"*. Given that your proposed metrics are motivated by the shortcomings of existing work, it is essential to faithfully represent those work and to identify actual, rather than imagined, flaws.
>
> Nonetheless, I consider this to be a relatively minor issue, one that can be addressed through some clarification, particularly when weighed against the validity of the proposed metrics.
>
>
>
> ## Validity of the metrics
>
> ```
> We validated our metrics in the original submission by showing 1) they capture behavior other metrics don't and 2) they correlate with detour performance for the taxi exercise (Table 2). We now include additional validation for Othello using detours.
> ```
>
> It's good to have multiple metrics, but it's also crucial that each one holds validity. I find it somewhat unclear how detour performance serves as a valid indicator in this context. Specifically, how can we be certain that the performance is solely influenced by the world model? For instance, could you clarify why $\text{Synthetic}$ appears to possess a near-perfect world model, whereas $\text{Championship}$ doesn't? To me, it's a typical phenomenon of exposure bias rather than a specific issue of implicit world model. $\text{Championship}$ is trained on expert data, where the prefix trajectories fall within a rather limited distribution. In contrast, $\text{Synthetic}$ is trained on randomly generated legal trajectories, which are distributed more evenly. As a result, the disparity between training and testing is significantly smaller for $\text{Synthetic}$ than for $\text{Championship}$.
>
>
> ```
> Common decoding strategies do provide mechanisms for determining sequence acceptance, like top-k, top-p, and threshold-based sampling (i.e. epsilon sampling)
> ```
>
> I respectfully disagree.  It seems there might be some conflation of concepts here. DFA are designed to accept or reject sequences through checking transition rules. In contrast, auto-regressive generative models don't have such mechanisms. All they have are token-level distributions and they are not trained to accept all "legal" sequences. Although it's possible to implement post-processing functions to pull an acceptance label out of it, such functions can influence metric scores, potentially weakening the soundness of the evaluation. Particularly, it's well known that the commonly used sampling-based decoding methods, e.g. top-p sampling, tend to select sequences that diverge significantly from the generative models' modes.
>
> I'm looking at Table 2 (top-k sampling) in the new pdf. With k=1, the scores for $\text{Random walks}$ are 0.40, 0.69, and 0.30, respectively. However, when k=4, these values rise to 0.64, 0.98, and 0.51. The disparity is quite pronounced to me.
>
> When it comes to Table 3 (top-p sampling), the compression precision for $\text{Random walks}$ is  0.16 when p=0.9 but 0.73 when p=0.99. The results somehow confirm my concerns.

---

> > ### Author Response · Authors · 2024-08-12
> >
> > Thank you for engaging with our paper. See our responses below.
> >
> > > _Summary of existing work_
> >
> > Thank you for clarifying your initial comment: _“Li et al. [1] do not solely rely on the validity of the top model predictions to assess the implicit world model. Instead, they directly probe the internal world state from the model, which has been the common practice of existing work ([2], [3]).”_
> >
> > We're glad it's clear that our paper not only describes the probe test of Li et al. [1] but also devotes empirical work to it.
> >
> > > _I find it somewhat unclear how detour performance serves as a valid indicator in this context. Specifically, how can we be certain that the performance is solely influenced by the world model?_
> >
> > This is a good question. We'd like to clarify a potential confusion about our paper: our metrics assess whether a model has a world model by assessing the _outputs_ of the model. In contrast, tests like the probe test do not test world models via their outputs, only their mechanisms (e.g. whether a representation encodes state and/or how a model uses this representation). This is why we and Li et al [1] include both kinds of metrics.
> >
> > The detours exercise assesses the outputs of a model. **A model whose output is consistent with the true world model will perform well on detours**; the detours exercise passes valid inputs to the model and sees if it can complete them successfully. In practice, we find that the detours exercise amplifies world model errors. The Championship Othello accuracy falls to 1% when detours are common, behavior that is not consistent with a model with the true world model. Our hypothesis as to why the Championship Othello doesn't have the correct world model is similar to your intuition -- that the way it's trained prevents it from differentiating between invalid moves and very bad moves. But our paper is focused on measuring whether a model behaves like the true world model rather than the reason as to why or why not.
> >
> > > _DFA are designed to accept or reject sequences through checking transition rules. In contrast, auto-regressive generative models don't have such mechanisms._
> >
> > Our metrics test whether a model's behavior obeys ground-truth rules. To implement these metrics, we need to define what it means for a model to accept or reject a sequence. This isn't unique to our setting; one way that Li et al [1] and Toshniwal et al. [2] test world models is by looking at whether the top-ranked prediction of a model is legal. Of course, a model generates more than one top-ranked token, and our different metrics and ablations provide different specifications.
> >
> > The results you point out are not contradictory. If a model has the true world model, all metrics (compression precision and distinction precision/recall) will score 100%. Conversely, if any metric is less than 100%, it won't have a perfect world model. As you note, once the metrics aren't perfect, their performance can differ and vary with a threshold parameter. To give a simple example: it's easy to ace compression (by saying every sequence has the same continuations) but then distinction suffers.
> >
> > This is similar to supervised learning: sensitivity and recall are all the same for a perfect classifier, but can differ for imperfect models in ways that vary with a thresholding parameter. The dependence on a thresholding parameter is not a weakness; it's crucial for decomposing error into precision and recall and measuring AUC. And our results across thresholding parameters and sampling mechanisms point to the same final conclusions.
> >
> > Thanks again for engaging with our paper. We'll update the paper in the revision to make these points more clear.
> >
> > [1] https://arxiv.org/abs/2210.13382
> > [2] https://arxiv.org/abs/2102.13249

---

> ### Comment · Reviewer_RjHx · 2024-08-13
>
> Thank you for your follow-up response. It helped me sort out some disagreements between us.
>
>
> I believe the root cause is this: "our metrics assess whether a model has a world model by assessing the outputs of the model. " While you state that you're evaluating the "implicit world model" of generative models, it seems to me that the outputs—and therefore the metrics—are influenced by various factors (such as the entropy of the learned distribution, decoding hyper-parameters, and algorithms) beyond just the implicit world model.
>
> ```
> The detours exercise assesses the outputs of a model. A model whose output is consistent with the true world model will perform well on detours
> ```
>
> I agree. But I'm not sure the model performs poorly on detours/proposed metrics if and only if it's inconsistent with the true world model. Even if there is some inconsistency, it might be exaggerated—or maybe even minimized; I'm not sure. The new results you presented earlier clearly confirm this concern.
>
>
> ```
> And our results across thresholding parameters and sampling mechanisms point to the same final conclusions.
> ```
>
> To be frank, I'm not sure I fully understand your point. If you're suggesting that the relative ranking of Random-walks > Shortest-paths > Noisy-shortest-paths is consistent across different decoding methods/hyperparameters, that's accurate. However, since you're proposing evaluation metrics, they should be applicable to more realistic models, not just those trained artificially. What happens if two models are trained on the same dataset, and their performance gap is much smaller? How can we ensure their relative ranking will still hold? This level of hyperparameter sensitivity raises concerns about the reliability of these metrics.
>
> On a related note, I have another question: why does Shortest-paths outperform Noisy-shortest-paths? If I understood correctly, Noisy-shortest-paths should involve more random exploration during training compared to Shortest-paths.
>
> ```
> Our hypothesis as to why the Championship Othello doesn't have the correct world model is similar to your intuition -- that the way it's trained prevents it from differentiating between invalid moves and very bad moves
> ```
> Indeed the gap between Championship and Othello in terms of differentiating between invalid moves and very bad moves are captured by the probe tests in Li et al. (2022). As shown in their Table 2, the error rate with Synthetic is significantly lower than with Championship.
>
>
> ```
> To implement these metrics, we need to define what it means for a model to accept or reject a sequence. This isn't unique to our setting; one way that Li et al [1] and Toshniwal et al. [2] test world models is by looking at whether the top-ranked prediction of a model is legal.
> ```
>
>
> Your setting is indeed unique. Sequence generation models are typically trained to maximize the probability of ground-truth sequences, making the selection of the top-ranked prediction—essentially MAP estimation—consistent with their training. While it's possible to use heuristics to sample multiple sequences from the distribution and argue that these sequences are "accepted" by the distribution (though what it means for a sample to be "accepted" by a distribution is unclear to me), this approach of inference differs from how the models are trained. This inconsistency arises due to the evaluation method, so it's questionable to blame the models if they don't perform well under these conditions. Moreover, if the top-1 prediction is typically used for reasoning tasks that rely on underlying world models, why should we be concerned with the consistency of top-$k$ predictions?
>
> ```
> The dependence on a thresholding parameter is not a weakness; it's crucial for decomposing error into precision and recall and measuring AUC.
> ```
>
> I'm not quite sure how those decoding hyperparameters function similarly to a classification threshold. With classification thresholds, it's easy to predict how adjusting them will affect precision and recall, making it a useful tool for balancing the precision-recall tradeoff. But when it comes to those hyperparameters and decoding algorithms, how exactly do they influence the metric values? For example, why does top-p decoding (p=0.999) result in a higher metric value for Random-walks compared to top-k decoding (k=4), yet performs worse for Shortest-path?

---

> ### Author Response · Authors · 2024-08-13
>
> Thanks for continuing to engage with our paper. We appreciate your feedback and will clarify our paper.
>
> > _I believe the root cause is this: "our metrics assess whether a model has a world model by assessing the outputs of the model."_
>
> We think a helpful comparison for our metrics is the next-token test performed by Li et al [1] and Toshniwal et al. [2]. Like our metrics, this test seeks to evaluate world models by the _behavior of their outputs_. Once we've established that a model behaves like it has the true world model, we can ask questions about the mechanism for why it behaves like that (e.g. via probing). For example, only after showing that OthelloGPT performs well on the next-token test do Li et al. [1] perform probing tests. The goal of our metrics is to test whether a model, when using common decoding schemes (we evaluate three different schemes), behaves like it has the true world model. For both our metrics and the next-token test, if a model's generations are influenced by factors like entropy, it would show up when practitioners decode from the model, and is therefore crucial for incorporating into metrics that assess world model behavior.
>
> At a high level, we believe that metrics which evaluate a model in a black-box manner, just based on observing their generative behavior, are vitally important and qualitatively distinct from those which rely on a certain implementation and/or internal parameters (we agree these are also useful). Several of the benefits of black-box metrics are the following
>  - They directly measure the quality of the actual thing we care about: the sequences generated by the model.
>  - They can be applied to any language generator, regardless of implementation. This also means that humans can be evaluated by the same measurements to get a baseline to compare models against.
>  - They provide a definition of what it means to have recovered a world model which is generic.
>
> We believe that the question “does a language model have an implicit world model?” can and should be evaluated by metrics that look only at the outputs of the model. Prior works [1, 2] introduced the ingenious idea of testing this question on rules-based systems (i.e. games and other types of DFAs) where we know the true world model. Our interpretation of having a world model corresponds to recovering the state space of the system. Automata theory tells us that this is the correct way of understanding these systems. **Recovering the true states of the world corresponds to an intuitive notion of capturing the world model.** The key idea is using this connection to automata theory to understand how well states are captured just by looking at sequences. We believe this connection as well as the extensive evaluation on new and standard tasks for interrogating world models contributes an important contribution to this line of work.
>
> >_I agree. But I'm not sure the model performs poorly on detours/proposed metrics if and only if it's inconsistent with the true world model. Even if there is some inconsistency, it might be exaggerated—or maybe even minimized_
>
> We're glad you agree that models whose outputs are consistent with the true world model will perform well on detours. Then we have the following:
> - If a model performs poorly on detours it doesn't behave like the true world model
> - Models like Synthetic Othello perform poorly on detours
> - This means they don't behave like the true world models
> - Previously proposed metrics suggest that all models we consider (including Synthetic Othello) do behave like correct world models
> - Our metrics correctly measure that they don't.
>
> This is a contribution of our work: we not only show that previously proposed metrics can lead to incorrect conclusions about world models, we also propose and validate new ones. We'd be happy to discuss any aspects of this further if you have questions or would like additional clarification.
>
> We agree that while poor performance on detours implies an incorrect world model, an incorrect world model can still perform well on detours. This is why it's important to have metrics that fully measure world model structure, like the ones we propose. Our detours exercise serves as one-way validation, which is the direction of disagreement between our proposed metrics and existing ones.

---

> ### Author Response · Authors · 2024-08-13
>
> > _However, since you're proposing evaluation metrics, they should be applicable to more realistic models, not just those trained artificially. What happens if two models are trained on the same dataset, and their performance gap is much smaller?_
>
> Like other papers about world model metrics [1, 2], we validate our metrics on models trained on artificial data. It is theoretically possible for model performance to depend on the decoding mechanism. It could imply that the models are so close in the amount of world model structure they capture that their performance depends on the decoding mechanism.
>
> You're correct that noisy shortest paths would involve more random exploration during training compared to shortest paths. It's a good question why it performs worse. One possibility is that it reaches a middle ground between random and constrained exploration that means it can't reap the benefits of either extreme. While we focus on world model metrics, it's important for future work to assess _why_ some models have world models and others do not
>
> > _Indeed the gap between Championship and Othello in terms of differentiating between invalid moves and very bad moves are captured by the probe tests in Li et al. (2022). As shown in their Table 2, the error rate with Synthetic is significantly lower than with Championship._
>
> The best probe accuracy for the Championship model from Li et al. [1] is indeed lower than the best accuracy for the Synthetic model (90.6% vs 98.3%). However a challenge of probing tests is the difficulty of distinguishing between two high scores [3, 4, 5] (Li et al [1] make no difference in their conclusions for each model's world modeling capabilities). In contrast, our metrics clearly differentiate the two Othello models (e.g. 0.98 compression precision for Synthetic and 0.00 for Championship) which is validated on detours (e.g. 99% accuracy for Synthetic with 0.50 detours compared to 1% accuracy for Championship).
>
> > _Sequence generation models are typically trained to maximize the probability of ground-truth sequences, making the selection of the top-ranked prediction—essentially MAP estimation—consistent with their training_
>
> Note that this is defining an acceptance criterion: top-k with k=1. Our metrics apply to this setting and we included experiments based on them. We also tested multiple ablations and two other sampling mechanisms to capture the diversity of ways in which it is possible to decode from LLMs. We believe it is important to consider various settings instead of constrain ourselves to a single one.
>
> > _Moreover, if the top-1 prediction is typically used for reasoning tasks that rely on underlying world models, why should we be concerned with the consistency of top-k predictions?_
>
> We performed multiple top-k ablations because of your suggestion to consider top-k sampling. This was a great suggestion: Even if top-1 predictions are typically used for reasoning, it certainly isn't the only sampling mechanism, and our experiments show the robustness to sampling mechanisms.
>
> > _But when it comes to those hyperparameters and decoding algorithms, how exactly do they influence the metric values? For example, why does top-p decoding (p=0.999) result in a higher metric value for Random-walks compared to top-k decoding (k=4), yet performs worse for Shortest-path?_
>
> There are patterns for how performance varies with threshold. For example, compression precision will typically increase as more sequences are accepted. This is because as more sequences are accepted, it is more likely that a model will accept a suffix for two prefixes that lead to the same state. While it's not an exact 1-1 mapping, we find this to be true for almost all models we consider.
>
> While we included top-k with the k=4 ablation for completeness, it is not an especially useful decoding mechanism for navigation. This is because there are only 8 possible cardinal directions, not all of which will be legal moves, and so decoding with top-k (k=4) can force the model to select sequences that aren't legal.
>
> [1] https://arxiv.org/abs/2210.13382
> [2] https://arxiv.org/abs/2102.13249
> [3] https://arxiv.org/abs/2102.12452
> [4] https://arxiv.org/abs/1805.01070
> [5] https://arxiv.org/abs/2004.03061

---

> ### Comment · Reviewer_RjHx · 2024-08-13
>
> Dear author,
>
> I have some quick questions.
>
> > *The best probe accuracy for the Championship model from Li et al. [1] is indeed lower than the best accuracy for the Synthetic model (90.6% vs 98.3%).*
>
> Could you direct me to the specific paragraph or section where this result is reported?
>
> > *While we included top-k with the k=4 ablation for completeness, it is not an especially useful decoding mechanism for navigation.*
>
> But nearly all models achieve their best metric scores in this setting. No?
>
> I will reply to other responses later.

---

> > ### Author Response · Authors · 2024-08-13
> >
> > Thank you for continuing to engage with our paper. We appreciate the continued discussion.
> >
> > > _Could you direct me to the specific paragraph or section where this result is reported?_
> >
> > These results are from Table 2 in [1]. The lowest probe error is 9.4% for Championship (layer 4 representation) and 1.7% for Synthetic (layer 7 representation). This corresponds to 90.6% accuracy for Championship and 98.3% for Synthetic.
> >
> > > _But nearly all models achieve their best metric scores in this setting. No?_
> >
> > Most models have better precision metrics when k=4 than lower k because of the intuition described in the last response: as more sequences are accepted, a model will accept more suffixes, meaning that precision typically improves. Recall has the opposite intuition. Note these are not strict 1-1 mappings, e.g. the recall for the Random Walks model gets worse for k=4 but better for the Shortest Paths model. We implemented the top-k metrics thanks to your suggestion and think they're useful to include in the paper, but the sensitivity of top-k metrics to the number of legal moves is why we prefer epsilon-based and top-p sampling.
> >
> > > _I will reply to other responses later._
> >
> > Thanks again for your engagement. Please let us know if you have any other questions.
> >
> > [1] https://arxiv.org/abs/2210.13382

---

> > > ### Comment · Reviewer_RjHx · 2024-08-14
> > >
> > > > Like other papers about world model metrics [1, 2], we validate our metrics on models trained on artificial data.
> > >
> > > Their papers don't propose new metrics but rather use chess and Othello as testbeds to probe the implicit world models of Transformers. The findings they gain from these testbeds are their primary contributions. In contrast, the main contribution of your manuscript lies in the newly proposed metric. A useful metric should differentiate between relatively good and relatively bad models—common scenarios in real-world applications—instead of merely distinguishing between the extremes of obviously good (training & testing in the same domain) and obviously bad (serious exposure bias).
> > >
> > > > These results are from Table 2 in [1]. The lowest probe error is 9.4% for Championship (layer 4 representation) and 1.7% for Synthetic (layer 7 representation). This corresponds to 90.6% accuracy for Championship and 98.3% for Synthetic.
> > >
> > > Thanks for the clarification. It seems we're on the same page: their probing test has identified the difference between Synthetic and Championship, though with less magnitude than yours.
> > >
> > > > We performed multiple top-k ablations because of your suggestion to consider top-k sampling. This was a great suggestion: Even if top-1 predictions are typically used for reasoning, it certainly isn't the only sampling mechanism, and our experiments show the robustness to sampling mechanisms.
> > >
> > > This isn't quite my question. What I’m really asking is: if we only use the top-ranked prediction in practice, why should we be concerned with other "accepted" sequences? Especially since acceptance is defined post-hoc, rather than being an intrinsic characteristic of the models.

---

> > > > ### Author Response · Authors · 2024-08-14
> > > >
> > > > Thank you for continuing to engage with our paper.
> > > >
> > > > > _The findings they gain from these testbeds are their primary contributions. In contrast, the main contribution of your manuscript lies in the newly proposed metric_
> > > >
> > > > Our paper not only proposes new metrics but also provides new findings about world models in testbed settings: we demonstrate that models previously thought to have correct world models, based on existing metrics, actually do not.
> > > >
> > > > > _A useful metric should differentiate between relatively good and relatively bad models—common scenarios in real-world applications—instead of merely distinguishing between the extremes of obviously good (training & testing in the same domain) and obviously bad (serious exposure bias)._
> > > >
> > > > We agree. We'd like to clarify a potential confusion: It's not true that our metrics can only be used to compare obviously good and obviously bad models. They are not arbitrary scores; they are grounded in a theoretically-based formulation of what it means to learn the structure of the world model. Consider the Random Walks model; it is good enough to pass the distinction test and performs better than the other models when detours are introduced, but still bad enough that its map is incoherent to us and it eventually fails when enough detours are added. Our metrics capture the difference between the Random Walks model and the other models trained on taxi data.
> > > >
> > > > Additionally, in addition to our experiments on models trained in synthetic testbeds, we devote empirical work to applying our metrics to 8 commonly used LLMs (Figure 4).
> > > >
> > > > > _If we only use the top-ranked prediction in practice, why should we be concerned with other "accepted" sequences?_
> > > >
> > > > If the only way a practitioner generates from an LLM is by using its top-1 prediction, then you're absolutely correct that they should only use the top-1 mechanism for our metrics (as provided in the rebuttal). But in practice there are many different ways people can decode from LLMs, each of which provide different notions of acceptance and rejection (see e.g. [1] for common decoding strategies). Our paper considers these to capture the diversity of ways in which practitioners decode from LLMs.
> > > >
> > > > [1] https://arxiv.org/abs/2402.06925

---

### Official Review · Reviewer_7KXV · 2024-07-12

**Soundness:** 4
**Presentation:** 4
**Contribution:** 4
**Rating:** 9
**Confidence:** 4

**Summary:**

This article proposes an evaluation framework for understanding whether or not transformers have learned an implicit world model. Existing metrics focus on next-token prediction and state probes, while this article proposes metrics inspired by the Myhill-Nerode theorem: Whether the network treats two action sequences arriving at the same state as having the same continuations, and whether the network properly distinguishes whether two states differ in allowable subsequent action sequences. They find that across large models trained on taxi data, Othello moves, and logic puzzles, these new metrics reveal far more weaknesses than previous metrics.

**Strengths:**

This is a brilliant article. Given the influence of LLMs, the question of whether strong next-token prediction leads to neural networks with emergent, implicit world models is open and very important. In my view, this article is the most convincing evidence yet with regards to this question. I can see this article having major impact.

Particular strengths include
- Excellent writing and presentation
- Choice of Taxi example
- Extensions to Othello and Logic puzzle domains
- Overwhelmingly clear results

**Weaknesses:**

I don't see real weaknesses. I have a couple minor suggestions for clarification:
- Figure 2 is a clever illustration but it could use some more detail in the caption, and I'm not sure I fully understand it. For the compression test, why are the only errors under the pink line, as opposed to covering other nodes in the graph on one side of the boundary? I have a similar question for the right hand side.
- Ideally the metrics in the tables, "Compression precision", "Distinction precision", "Distinction recall", would be more clearly defined somewhere in bold, like other terms in the article.

**Questions:**

- I'm a little puzzled why the models have low distinction precision/recall. If the two states they are meant to distinguish are sampled randomly, why would the model confuse their continuations? Some more intuition here--especially providing example errors-- would be helpful.

- How much training was provided to the models?

**Limitations:**

I see no issues here.

---

> ### Author Rebuttal · Authors · 2024-08-07
>
> Thank you for your thorough and insightful review of our paper. We appreciate your enthusiasm for the work and your remarks on the quality and significance of our paper.
>
> > _Figure 2 is a clever illustration but it could use some more detail in the caption, and I'm not sure I fully understand it. For the compression test, why are the only errors under the pink line, as opposed to covering other nodes in the graph on one side of the boundary? I have a similar question for the right hand side._
>
> This is a good point. We apologize for the confusion. The figure was intended to illustrate boundary errors, but you're right that examples to the side of the boundary are also errors. We've updated the figure and caption to make more clear that we're just showing boundary errors and we've also made a few stylistic updates as well (see the shared PDF response).
>
> > _Ideally the metrics in the tables, "Compression precision", "Distinction precision", "Distinction recall", would be more clearly defined somewhere in bold, like other terms in the article._
>
> Thank you for the suggestion. We'll add these definitions in bold in the main text.
>
> > _I'm a little puzzled why the models have low distinction precision/recall. If the two states they are meant to distinguish are sampled randomly, why would the model confuse their continuations? Some more intuition here--especially providing example errors-- would be helpful._
>
> This is a good question. Your intuition for the distinction metric is right. An important point is that the distinction metric tests both whether the states are correctly distinguished and whether a model produces the correct difference in continuations. In other words, to perform well at the distinction test, a model needs to know the exact set of continuations that are legal in one state but not the other. One reason this could be hard is if the Myhill-Nerode boundary between states is large; then models need to correctly differentiate potentially long-range and complex continuations.
>
> In the navigation setting, we found two kinds of interpretable distinction errors. One is that intersections that are geographically close to each other are confused for one another. Another is that intersections on streets with the same traffic patterns (e.g. the intersecting streets are one-way in the same directions) are also confused for each other. We found other kinds of distinction errors as well that were harder to interpret; an interesting research question is understanding why models fail to distinguish states that humans can distinguish easily. We'll add qualitative examples to our updated revision and we thank you for suggesting them -- they'll make the paper stronger.
>
>  > _How much training was provided to the models?_
>
> Training sizes:
> - Shortest paths: 2.9M sequences (120M tokens)
> - Noisy shortest paths: 31M sequences (1.7B tokens)
> - Random walks: 91M sequences (4.7B tokens)
>
> We trained models until the validation loss increased or stopped improving. This ranged from ~12 hours for the smallest models and datasets (shortest paths) and 48 hours for the largest models (random walks) on 8 A100 GPUs. We'll update the paper to make these training details more clear.

---

> > ### Comment · Reviewer_7KXV · 2024-08-07
> >
> > Thanks for your reply. I reaffirm my very positive assessment of the work!

---

### Official Review · Reviewer_YkFb · 2024-07-12

**Soundness:** 3
**Presentation:** 3
**Contribution:** 3
**Rating:** 5
**Confidence:** 1

**Summary:**

This paper proposes new evaluation metrics to assess whether a learnt world model is indeed learning the underlying dynamics or logical reasoning required to fully decipher a new domain. The paper sheds light into how world models should be evaluated, compared to what is being done in the literature currently and finds that current metrics can lead to instability or lacks enough evidence that a true underlying world model of the domain is indeed learnt.

**Strengths:**

The core idea of the work seems interesting; but unfortunately as a review I do not have the necessary background to fully or properly assess this work.

This work tries to address an important question for how current world models should be evaluated, and uses literature from automation theory to propose new metrics to see if the underlying logic of the domain can indeed be learnt by the world model.

My assessment of the work is rather high level with not enough technical background to assess the core contributions of this work; however, if this wiork is technically sound and can be made addressable with enough intuitions to general audience not familiar with automata theory, this work can indeed be quite significant and important to the community.

**Weaknesses:**

My only comment would be that the paper should perhaps provide more background and intuition from automata theory, to justify the tools that are introduced to be able to fully understand this work. The core idea seems interesting but the experimental and contributions of the work, if it can be explianed more generally, would perhaps be more useful.

**Questions:**

...

**Limitations:**

....

---

> ### Author Rebuttal · Authors · 2024-08-07
>
> Thank you for your careful and insightful review of our paper. We're glad that you think the paper is addressing an important question and that it has the potential to be "quite significant and important to the community".
>
>
> > _My only comment would be that the paper should perhaps provide more background and intuition from automata theory, to justify the tools that are introduced to be able to fully understand this work. The core idea seems interesting but the experimental and contributions of the work, if it can be explained more generally, would perhaps be more useful._
>
> You bring up a great point. We'll use the extra space available to us in the camera-ready revision to move the definition of DFAs from the appendix to the main text. We'll also provide the following high-level explainer:
>
> At a high level, a DFA is a collection of states and a set of rules that govern how states are related to each other. For example, consider the game of Othello. This is a DFA where each state is essentially a different board position. The rules are similar to the rules of Othello; they tell you not only which moves are legal, but also how playing a particular move at a given board takes you to a new board. While we use Othello here as an illustrative example, DFAs are general: for example, navigation is a DFA (where the true state is which intersection you're at). They're common in other application areas, such as search engines, control systems, and genetics.
>
> DFAs are commonly used to study world models because they let us compare a model's predictions to the true rules [1, 2, 3, 4]. One popular method is the next-token test: given a sequence that corresponds to a true state, is the model's top-predicted next token valid under the rules of the DFA at that state? For example, given a sequence of Othello moves (encoded as tiles, e.g. "34 12 30 23 26"), is a transformer's prediction of the next tile a legal move?
>
> Our paper shows this next-token test can be misleading; models that are very far from having the true world model can perform very well at this test. A classic result -- the Myhill Nerode theorem -- provides intuition as to why: states aren't only defined by which individual next actions are legal, but rather by which (potentially long) sequences of actions are legal. For example, there are many Othello boards that have the same set of legal next moves, but the _sequences_ of legal moves differ when we consider longer sequences. As a result, testing whether the true world model has been recovered requires going beyond next-token tests.
>
> This motivates two new properties a model must satisfy if it has recovered the true world model recovery:
> - **Sequence compression:** if two sequences lead to the same state, a model shouldn't distinguish them.
> - **Sequence distinction:** if two sequences lead to distinct states, a model should distinguish them.
>
> Our metrics directly measure these properties. Sequence compression is evaluated by sampling two sequence prefixes that lead to the same state and making sure a model's predicted continuations of those sequences are similar. For Othello, this corresponds to checking whether a model's outputs are similar for two move sequences that result in the same board. Our measure of sequence distinction is similar: it's evaluated by sampling two sequence prefixes that lead to different states and making sure that a model's predicted continuations reflect the differences in the continuations. In other words, this metric tests how well a model captures how different board positions are different.
>
> Empirically, these metrics capture important aspects of world model recovery that other metrics do not. In the taxi example, prior metrics for assessing world models would lead us to conclude that transformers trained on taxi rides have world models. But our metrics come to a different conclusion: these models are far from recovering the true world model. We validate this by recovering the implicit map of NYC and showing it's nonsensical, along with showing that each model's navigation performance breaks down when detours are introduced. Our metrics also discern between two different types of Othello models: the model trained on Synthetic games has a world model, while the model trained on Championship games does not by our metrics. We validate this with a similar detours exercise (see the PDF for more details), where the Synthetic model produces near-perfect games regardless of detours and the Championship model fails immediately. There is a clear distinction between Championship and Synthetic models, but this is only captured by our metrics; existing metrics would lead us to conclude that both have world models.
>
> We hope this is helpful. We've also included more motivation for including DFAs in our response to Reviewer S5LR. Please let us know if anything is unclear. If these comments have addressed your concerns, we hope you'd consider raising your score.
>
> [1] https://arxiv.org/abs/2106.00737
> [2] https://arxiv.org/abs/2210.13382
> [3] https://arxiv.org/abs/2102.13249
> [4] https://arxiv.org/abs/2310.07582

---

> > ### Author Response · Authors · 2024-08-12
> >
> > Thank you again for taking the time to review our paper. We appreciate your comments and believe the paper will be stronger because of them.
> >
> > We were wondering if you had any more questions our review didn’t address. Since the discussion period ends tomorrow we want to make sure we have time to address your points. If you don’t have any more questions, we hope you’d consider changing your score.

---

### Author Rebuttal · Authors · 2024-08-07

We thank the reviewers for their careful evaluation and feedback. We're glad you found our paper "brilliant" (7KXV) and offering a "novel perspective" (RjHx), with the potential to be "quite significant and important to the community" (YkfB) and to have "major impact" (7KXV). Moreover we appreciated your comments on the strength of the empirical evidence (7KXV, RjHx, S5LR) and quality of the writing (7KXV, RjHX, S5LR).

In response to the reviewers’ suggestions, we've included updated results in the attached PDF with:

- **Additional validation results:** Our original submission validated our metrics empirically using maps and detours for the navigation data. We now include a similar empirical validation for Othello. In our original submission, we showed that while existing metrics conclude that both Synthetic and Championship Othello models recover the true world models, our metrics find that only the Synthetic model recovers the true world. We include an additional "detour" exercise  (where a model's predicted move is replaced with another legal one) to validate this discernment for Othello. While the Synthetic model produces near-perfect games regardless of detours, the Championship model fails immediately. A model that recovers the true world model will succeed regardless of detours. There is a clear distinction between Championship and Synthetic models, but this is only captured by our metrics; existing metrics would lead us to conclude that both have world models.

- **Additional acceptance criteria:** While our submission considered fixed probability thresholds to define transformer acceptance, an advantage of our framework is that it can be implemented with different acceptance criteria. In response to Reviewer RjHx's suggestion, we've also added results for top-p and top-k sampling (along with ablations of p and k). These come to the same conclusions as our original criterion, which, along with the ablations in Table 3 of our submission, show the robustness of our metrics to acceptance criteria.

- **Updated figure:** In response to Reviewer 7KXV's suggestion, we've updated Figure 2 from our original submission to make the interpretation more clear.

These results will make the paper stronger, and we thank you for suggesting them. We’ve also included more details about these results in our individual responses to each reviewer.

---

### Decision · Program_Chairs · 2024-09-25

**Decision:**

Accept (spotlight)

**Comment:**

This paper created a robust discussion between authors and reviewers. After carefully reading the paper and the ensuing discussion, I recommend acceptance.

The problem of recovering world models from sequence data is important, and assessing whether or not an LLM has indeed captured such a model is a problem of significant interest in the community. This paper proposes a set of theoretically motivated tests that have more discriminative power than other model-agnostic tests; the authors demonstrate the power of these tests on several interesting problems.  As such, this paper provides an important contribution to the discussion on interpretability.  Furthermore, the paper is very well-written, and has a nice mix of theoretical ideas and practical experiments.

The discussion among the reviewers and the authors hinged on the theoretical underpinnings of these tests, and their properties.  On the theoretical side, the concern raised by the reviewers pointed out difficulties in interpretation of the tests and the essential "type error" between DFAs that accept sequences and generative models that produce things according to probabilities. While I share the type error concern to some extent, I find that the practical utility of the tests, combined with the clarity of the discussion and the issues that the paper raises, outweighs this concern.  It may very well be that these tests need more theoretical refinement, but I am confident that the community will engage these ideas and provide it.  Said another way: I do not believe anything about the proposed tests is /wrong/; I also find that the tests are intrinsically useful and motivated enough to stand independent of any interpretation we may choose to project upon them. The reviewers also had some concerns about the properties of these tests, and whether or not things like relative rankings are consistent.  While we can always demand more theory, more perfection, more power, and more utility, I find that the contribution as it stands is more than sufficient to warrant publication in NeurIPS.